

# A Bias Correction Scheme for FY-3E/ HIRAS-II Observation Data Assimilation

Hongtao Chen[1], and Li Guan[1*]

[1] *China Meteorological Administration Aerosol-Cloud and Precipitation Key Laboratory, Nanjing University of Information Science and Technology, Nanjing 210044, China.*

Correspondence to: Li Guan (liguan@nuist.edu.cn)

**ABSTRACT—Meteorological satellite data have been extensively utilized in global numerical weather prediction systems and have a**
**positive impact to improve forecast accuracy. In order to correctly assimilate satellite radiance observations in data assimilation systems,**
**the systematic observation biases must be corrected to conform to a Gaussian normal distribution with a mean of 0. By selecting**
**appropriate air-mass predictors through correlation assessment, a two-step bias correction scheme (namely the scan-angle bias**
**correction and the air-mass bias correction) is established in this paper based on radiation observations of FY-3E/ HIRAS-II from 1 to**
**31 January 2023. The results indicate that FY-3E/HIRAS-II O-B (observation-simulation) bias exhibits scanning angle bias dependence**
**from nadir to limb field of view. Statistics have found that this scanning angle bias does not depend on latitude band. After scan-angle**
**bias correction using statistical scan-angle correction coefficients, the dependence of the O-B biases on the scan angle can be eliminated.**
**The second step is to perform air-mass correction. Our correction scheme is compared with the air-mass bias correction scheme in**
**NCEP-GSI. Although the scan angle influence is also considered in NECP-GSI scheme, it does not account for the water vapor effect in**
**the atmosphere. Consequently, the correction effect is not good for channels with lower peak height of weighting function, resulting in a**
**slightly residual positive bias after correction. The combination of air-mass predictors (model surface skin temperature, model total**
**column water vapor, thickness of 1000-300 hPa, and thickness of 200-50 hPa) selected through importance assessment in this study**
**effectively eliminates the air-mass biases. The systematic biases between observed brightness temperature and background simulated**
**brightness temperature from background atmospheric field for all HIRAS-II channels significantly decrease after bias correction, and**
**the bias distribution essentially follows a Gaussian normal distribution with a mean of 0. The bias correction scheme has a significant**
**improvement for the analysis at upper air and near surface.**
**Index Terms—FY-3E/HIRAS-II, bias correction, data assimilation, numerical weather prediction (NWP)**

## 1.  INTRODUCTION

The quality of the numerical weather prediction (NWP) largely depends on the accuracy of the initial atmospheric conditions,
provided by the data assimilation system (Auligné et al., 2007b). Satellite data is an important input observation source to data
assimilation systems, it is of great significance to correctly assimilate satellite radiation data to improve the accuracy of the
numerical weather prediction (Zhang et al. 2023).
Satellite observations have the advantages of wide coverage and high temporal and spatial resolution, which greatly
complement the data gaps in areas lacking conventional observations worldwide. Moreover, satellite radiations in infrared or
microwave bands exhibit strong sensitivity to meteorological elements (e.g., temperature, humidity) within the atmospheric
structure (including the Earth's surface) (Li et al., 2019). Notably, satellite-borne infrared hyperspectral atmospheric sounders such
as AIRS (Atmospheric Infrared Sounder), CrIS (Cross-track Infrared Sounder) and IASI (Infrared Atmospheric Sounding
Interferometer) can obtain global meteorological observations with multiple channels and high vertical and spectral resolution.
These observations, particularly three-dimensional atmospheric temperature and humidity profiles, have been extensively utilized
in global numerical weather prediction with a significant positive impact. In contrast to other infrared hyperspectral atmospheric
sounders, HIRAS-II (Hyperspectral Infrared Atmospheric Sounder-II) is carried on FY-3E, the world's first early morning polar-



orbiting meteorological satellite. The satellite launched in 2021 effectively fills the time gap of satellite observations within a 6-
hour assimilation window, ensuring 100% coverage of satellite observations within the assimilation time window. As the world's
only infrared atmospheric sounder operating in an early morning polar orbit, it is imperative to assimilate its observations into data
assimilation systems (Zhang et al., 2024).
One of the most important steps in data assimilation is bias correction, especially for the bias correction of satellite observation
(Yin et al., 2020). Satellite observation currently account for the vast majority of all assimilated atmospheric observations, and
satellite observations have a strong influence on the quality of the main output meteorological parameters (e.g., temperature, winds
and humidity) from the analysis (including the reanalysis), as well as the additional information generated by the assimilating
forecast models (e.g., precipitation, cloudiness and radiative fluxes). During assimilating the meteorological satellite-observed
radiation data, the target functional atmospheric data assimilation method based on statistical optimal estimation requires the errors
of satellite observations (O) and background simulations (B) all to be an unbiased Gaussian distribution. Therefore, if uncorrected
satellite observations are directly absorbed by the data assimilation system, the accuracy of the analyzed fields will be affected,
thus destroying the global NWP system in a very short time (Dee., 2004). However, satellite observations (O) and background (B)
in fact are always systematically biased. The systematic bias between O and B may originate from observation errors of the
instrument itself, simulation errors of the fast radiative transfer model, forecast errors of the NWP system (as the input atmospheric
state parameters to radiative transfer models), errors introduced during data preprocessing steps, among others. The sources of
systematic bias are complex, but the bias can be quantitatively estimated by statistical $O - B$. If the unbiased assumption is not
valid, the deviation of the observation error $\mu_o = \overline{O - T}$ and the deviation of the simulation error $\mu_b = \overline{B - T}$ must be subtracted
from the observation and simulation brightness temperature, respectively. $T$ represents the true value of brightness temperature.
In the presence of errors in observation and background, $\mu_{O-B} = \overline{O - B}$, $\mu_{O-B}$ is the deviation between observed and simulated
brightness temperatures. This expression provides a basis for bias estimation and bias correction, so the value of $\mu_{O-B}$ can be
estimated even in the absence of the true value based on the statistical samples of $O - B$. In order to properly use satellite
observations in the assimilation system, the $O - B$ must first be corrected to produce an unbiased analysis (Zou., 2023).
McMillin et al. (1989) first proposed a bias correction scheme for the TIROS Operational Vertical Sounder (TOVS) using the
observed radiation from MSU (Microwave Sounder Unit) channels 2, 3 and 4 as air-mass predictors. Eyre (1992) adjusted the
scheme by incorporating cloud radiation, but the air-mass biases still remained. It is worth mentioning that Harris and Kelly (2001)
proposed a revolutionary bias correction scheme that divides the correction into two parts: the scan angle bias correction and the
air-mass bias correction. On one hand, the scheme incorporated the latitude dependency into scan angle bias correction, and on the
other hand, it replaced observation-based predictors variables with model background-based predictors. Although the scheme has
achieved remarkable progress, it is a static off-line scheme that relies on bias correction coefficients calculated in advance based
on historical samples and does not account for the evolution of biases with time and with weather systems. Subsequently, many
researchers developed variational bias correction schemes. Dee (2004) proposed a variational scheme for adaptive radiation bias
estimation and correction based on the ECMWF (European Centre for Medium-Range Weather Forecasts) assimilation system.
Zhu et al. (2014) objectively evaluated the effect of the variational correction scheme in NCEP (the National Centers for
Environmental Prediction) - GSI (Gridpoint Statistical Interpolation).
Due to the dependence of systematic biases on instruments (or sensors), many scholars have developed specific bias correction
schemes suitable for respective satellite instruments. Liu et al. (2007) proposed a bias correction scheme based on the radiation
data from the ATOVS (Advanced TIROS Operational Vertical Sounder) instruments onboard NOAA-15/16/17 polar-orbiting
meteorological satellites. Li et al.  (2016) proposed a bias correction scheme for IASI (Infrared Atmospheric Sounding
Interferometer) suitable for the GRAPES assimilation system, based on the approaches of Harris and Kelly. Li et al. (2019) assessed
the capability of air-mass predictors for bias correction in the NCEP GSI assimilation system using CrIS radiance data and proposed
an improved bias correction scheme based on the periodic characteristics of observation minus background biases. Yin et al. (2020)
evaluated the observation quality of the FY-4A/GIIRS (Geostationary Interferometric Infrared Sounder) longwave infrared



channels using GRAPES 4D-Var assimilation system and applied an off-line bias correction scheme to correct the O-B biases of
these channels.

Most of the above radiation bias correction studies mainly focus on spaceborne microwave radiometers, and there is limited
research on observation bias in infrared hyperspectral atmospheric sounders, particularly for the HIRAS-II onboard the early
morning polar-orbiting satellite FY-3E launched recently. Therefore, a bias correction scheme suitable for FY-3E/HIRAS-II is
established in this paper based on the selection of the optimal air-mass correction predictor combination using its radiation
observation from 1 to 31 January 2023. In addition, the scheme is compared with the air-mass correction scheme in NECP-GSI to
quantitatively evaluate its bias correction effect. Finally, the potential positive impact of the scheme to enhance the accuracy of
data assimilation is check based on a one-month assimilation experiment.

2.   DATA AND MODEL

This article investigates a bias correction study on the radiation observations from the hyperspectral infrared atmospheric
sounder HIRAS-II aboard the Fengyun-3E satellite. HIRAS-II is an interferometer Fourier transform spectrometer with continuous
coverage of infrared wavelength from 3.92 to 15.38 μm, consisting of 3041 channels with a spectral resolution of 0.625 cm$^{-1}$.
HIRAS-II measures Earth and atmosphere in the conventional mode through a cross-track rotary scan mirror that provides the scan
angles vary from −50.4° to +50.4°. Each scan line observes 32 fields of regard (FOR), including 28 contiguous ground targets,
2 cold spaces and 2 onboard blackbody targets. Each FOR consists of a 3×3 array of fields of view (FOVs) and the approximate
resolution for the nadir FOV is 14 km. A total of 31 days' HIRAS-II Level 1 radiation data from 1 to 31 January 2023 is used in
our research as the satellite observations (O). The HIRAS-II Level 1 radiation data obtained from the Fengyun Satellite Data
Center: http://data.nsmc.org.cn.

The National Centers for Environmental Prediction (NCEP) Global Forecast System (GFS) 6h forecast filed valid at 0000,
0600, 1200, and 1800 UTC, are used as input to the fast radiative transfer model. (Data are available at
https://rda.ucar.edu/datasets/ds0.841/). GFS data is a regular latitude-longitude grid data with a spatial resolution of 0.25°×0.25°
and the atmosphere is divided into 41 vertical layers from 1000 hPa to 0.01 hPa. The atmospheric state parameters include the
profile of temperature, humidity and ozone, et al.

The GFS data are spatialy-temporaly matched to HIRAS-II FOVs as follows: for each HIRAS-II field of view (FOV), perform
bilinear interpolation on GFS forecast value by selecting the 4 closest grid points. Since the samples collected in this study are
clear-sky observations over sea, the clear-sky atmosphere does not change much within 0 to 3 hours, so based on the observation
time of each HIRAS-II FOV, select the two spatially matched GFS data that are closest to HIRAS-II's observation time for linear
interpolation.

The fast radiative transfer model employed in this study is RTTOV v12.3. RTTOV is a widely used fast radiative transfer
model, suitable for satellite radiometers, spectrometers, and interferometers in visible, infrared and microwave bands. It can
simulate satellite-observed radiation based on the atmospheric and surface state vectors input by users (Saunders et al., 2018) The
spatially and temporally matched GFS data are used as input to calculate simulated radiation (B) for FY-3E/HIRAS-II.

3.   QUALITY CONTROL

To ensure the rationality and effectiveness of the HIRAS-II data used for statistics, a quality control process is performed
before calculating the bias correction coefficients. The following steps are included:
1)   Cloud detection
Cloud detection is a crucial step before satellite data assimilation. Since the infrared hyperspectral atmospheric sounder cannot
penetrate clouds during the measurement, its observations are highly susceptible to the influence of clouds and precipitation, and
simulations of the fast radiative transfer modes under cloudy conditions have considerable uncertainty, so selecting the clear-sky



field of view with high confidence can greatly reduce simulation errors in the fast radiative transfer models. In this study, the
temporally and spatially matched FY-4A/AGRI (Advanced Geostationary Radiation Imager) cloud mask products are employed
for FY-3E/HIRAS-II FOV cloud detection. The HIRAS-II has a coarse spatial resolution of 14 km at nadir, whereas the AGRI
cloud mask product has a higher spatial resolution of 4 km. Therefore, approximately 4×4 AGRI pixels are co-located within each
HIRAS-II FOV. These clear HIRAS-II FOVs are retained during statistics when all the spatially matched AGRI pixels within the
HIRAS-II FOV are identified as clear. The FY-4A/AGRI Level 2 4 km cloud mask product with 4 km resolution can be downloaded
from the Fengyun Satellite Data Center: http://data.nsmc.org.cn.
2) Surface type detection
The surface emissivity calculated by the fast radiative transfer model is relatively accurate when the underlying surface within
a satellite field of view is more uniform. The underlying surface over land is relatively complex, leading to some errors during
simulation, while ocean surfaces are relatively uniform. The surface type of each FOV is determined based on the FY-3E/HIRAS-
II Level 1 LSM (Land Sea Mask) product. Only these ocean satellite-observed scenes are kept during statistics.
3) Data Thinning
Each field of regard (FOR) of FY-3E/HIRAS-II consists of 3 × 3 arranged FOVs with 14 km spatial resolution. Only the
observations from the central fifth FOV within each FOR are retained, so the spatial resolution is approximately 45 km after
thinning.
4) Clear-sky detection using window channels
To further eliminate the influence of clouds, a threshold test based on the O-B biases at these window channels 925 cm$^{-1}$, 970
cm$^{-1}$, and 1111 cm$^{-1}$ is performed for each thinned FOV. The FOV with O-B bias exceeds -4 to 4 K at any channel is discarded.
5) Outlier detection
The FOV with observed brightness temperature in any channel exceeding the value range (150-350 K) will be discarded. If
the O-B biases in any channels exceed three times its standard deviation, the FOV is also discarded.
After the above quality control steps, a total of 67534 samples were counted. 34844 samples from 1 to 14 January 2023 are
used as the training data for fitting bias correction coefficients, and 32690 samples from 15 to 31 January 2023 are used as the
testing data to examine the correction effect.
## 4. BIAS CORRECTION SCHEME
HIRAS-II radiation bias correction is divided into two steps referring to the bias correction method of Harris and Kelly (2001)
in this study.
*4.1 Scan-angle bias correction*
The field of view is susceptible to deformation as the scan angle increases when the satellite sensor performs a cross-track
scanning to both sides of scan lines, which lead to unavoidable observation biases relative to the nadir FOV. For each channel,
calculate the global or regional mean bias of every scan position (angle) relative to the central nadir position:
$$S(\theta) = \bar{R}(\theta) - \bar{R}(\theta = 0) \tag{1}$$

Where $\bar{R}$ refers to the mean radiation at different scan angles,$\theta$ represents the scan position (scan angle) and $S$ denotes
the scan bias. A significant improvement made by Harris and Kelly (2001) to this scheme is incorporating the dependency of biases
on latitude band. The Earth is divided into 18 latitude bands with 10 degrees interval and the correction coefficient is computed
respectively. After a long period of lager sample size statistics, it has been shown that the scan biases of HIRAS-II do not exhibit
a pronounced dependence on latitude band as found in microwave instruments. Therefore, the influence of latitude is not considered
for the scan bias correction in this study.



*4.2 Air-mass bias correction*
The O-B biases generally exhibit variations associated with the properties of air-mass (and the surface) due to the inaccuracies
in the radiation calibration of satellite instruments, the error from the fast radiative transfer models and NWP systems. The air-
mass bias correction scheme primarily involves establishing a multivariate linear regression equation that relates the air-mass
predictors $x_i (\mathrm{i} = 1,2,\dots,\mathrm{n})$ to the air-mass biases, the air-mass biases $r_j$ for each channel $j$ can be calculated:
$$r_j = \sum_{i=1}^{n} a_{ji} x_i + c_j \tag{2}$$


Where $a_{ji}$ and $c_j$ are calculated by least square fitting using a large number of samples.
$$a_{ji} = \sum_{k=1}^{n} \langle D_j, x_k \rangle [\langle X, X \rangle]_{ki}^{-1} \tag{3}$$


Here,$\langle \dots, \dots \rangle$ represents covariance, $k$ is the sample number, $X$ is the vector of $x_k$, and $D_j$ denotes the O-B bias in
channel j.
The success of air-mass bias correction depends on the selection of air-mass predictors. The commonly used air-mass
predictors in the ECMWF and NCEP-GSI assimilation systems are shown in Table 1. Here, $p$ represents the air-mass predictors
in the ECMWF assimilation system and $p'$ represents the air-mass predictors in GSI. The predictors $p_1$-$p_7$ are related to the
atmospheric conditions within the satellite-observed scenes. These predictors primarily reflect the systematic errors in fast radiative
transfer models and NWP models. The predictors $p_8$-$p_{10}$ are employed to correct residual biases after scan bias correction (Dee
and Uppala., 2009). The Parameter thickness is calculated as
$$Pred_{thickness} = kth \times \sum_{i}^{N} tv(i) \times \ln P(i) \tag{4}$$

Here, $kth = gas_{constant}/gravity (gas_{constant} = 287.0\,K,\ gravity = 9.81\,N/kg)$, $N$ is atmosphere layers, $P$ is
atmospheric pressure, $tv$ is the parameter characterizing atmospheric temperature and humidity and calculated as
$$tv(i) = \frac{T(i)}{2} \times \left[1.0 + 0.608 \times \frac{q(i)}{2}\right] \tag{5}$$

Where, $T$ and $q$ represent RTM level temperatures and moistures, respectively.
In the NECP-GSI assimilation system, the predictor $p'_0$ represents a global constant offset. The predictor $p'_1$ is a function of
the satellite scan angle $\theta$ and is primarily used to correct residual scan biases. The predictor $p'_2$ is only used for the correction of
clear-sky microwave instrument radiation over the ocean to eliminate residual cloud interference. For non-microwave instruments,
the value of the predictor is set to 0. $p'_3$ and $p'_4$ is the predictor of "temperature lapse rate", $\Delta \tau$ and $\Delta T$ represents the vertical
variation rate of transmittance and temperature, respectively. The predictor $p'_4$ represents the convolution of $\Delta \tau$ and $\Delta T$, and
$p'_3$ is the square of the former. The predictor $p'_3$ and $p'_4$ reflect instrument and RTM errors. When the frequency of channel is
shifted or the spectral settings in the RTM are inaccurate, the calculated transmittance profile will move up/down in the atmosphere
(if the atmosphere is not isothermal). If the transmittance is moved up slightly, the weight function of the channel is shifted upwards
and the brightness temperature should be decreased if the temperature decreases with height. Conversely, the brightness
temperature will increase in the case of a temperature inversion. (Zhu et al., 2014).
Harris and Kelly (2001) as well as Liu et al. (2007) used four predictors (model surface skin temperature, model total column
water vapor, thickness of 1000-300 hPa and thickness of 200-50 hPa) as the optimal combination to the TOVS's air-mass bias
correction. Auligné et al. (2007a) employed a predictor combination of thickness of 1000-300 hPa, thickness of 200-50 hPa,
thickness of 50-5 hPa and 10-1 hPa to correct the air-mass biases in ATOVS. The predictor combinations used in the
above-mentioned studies are primarily designed for microwave instruments. The predictors used in the ECMWF for all infrared
hyperspectral instruments operating on polar-orbiting platforms are summarized in Table 2 (Auligné et al., 2007a; Collard and
McNally., 2009; Eresmaa et al., 2017). In order to evaluate the optimal combination for FY-3E/HIRAS-II, several typical channels
(737.5cm[-1], 900cm[-1], 1040cm[-1], 1279.375cm[-1], 1476.25cm[-1] and 1809.375cm[-1]) are taken as examples in our study. The importance



of predictors $p_1$-$p_6$ for FY-3E/HIRAS-II is evaluated based on the diagnostic scheme proposed by Auligné (2007a). Since we use
a two-step bias correction scheme, angle-related predictors are not involved in the assessment. This diagnostic scheme evaluates
the bias correction effect based on the ability of each predictor to reduce the root mean square error of radiation biases. The results
are shown in Figure 1. Figure 1(a) and (b) show the spectral positions and the weighting functions (WF) of each selected channel.
The wavenumber of HIRAS-II channel No.141, 401, 626, 1008, 1323 and 1855 are 737.5cm$^{-1}$, 900cm$^{-1}$, 1040cm$^{-1}$, 1279.375cm$^{-1}$,
1476.25cm$^{-1}$ and 1809.375cm$^{-1}$, respectively. The peak heights of weighting function are 535.2 hPa, 1070.9 hPa, 29.1 hPa, 852.8
hPa, 300 hPa and 500.2 hPa in sequence. The six channels correspond to the long-wave $CO_2$ absorption band, the long-wave
window channel, the $O_3$ absorption band, and the water vapor absorption bands at three different heights in turn. Figure 1(c)
indicates the importance assessment of different predictors in different channels, where the horizontal axis indicates the channel
No. and the vertical axis is the diagnostic coefficients. A higher coefficient indicates a stronger correlation. The different colored
bars represent different predictors. It can be seen from Figure 1 (c) that the $O_3$ channel 626 and the water vapor channel 1323 with
higher height of weight functions have a higher correlation in predictors thickness of 10-1 hPa and 50-5 hPa. However, other four
channels commonly used for data assimilation systems are strongly correlated with the model background surface skin temperature,
model total column water vapor, thickness of 1000-300 hPa, and thickness of 200-50 hPa. Data assimilation systems generally do
not assimilate strong $O_3$ absorption channels at present and the water vapor content in upper atmosphere is scarce, so a predictor
combination including model surface skin temperature, model total column water vapor, thickness of 1000-300 hPa and thickness
of 200-50 hPa is selected to correct the air-mass biases for HIRAS-II in this research. Furthermore, to assess the effectiveness of
our bias correction scheme a comparison is made in this study between the correction results of the NCEP-GSI predictor
combination ($p'_0$-$p'_4$) and the proposed method.

Table 1 Bias predictors implemented in ECMWF and GSI

| ECMWF | NCEP-GSI |
|---|---|
| $p_0$: 1 (constant) | $p'_0$: 1 |
| $p_1$: 1000-300 hPa thickness | |
| $p_2$: 200-50 hPa thickness | $p'_1$: $\frac{1}{10} \times (\frac{1}{\cos\theta} - 1)^2 - 0.015$ |
| $p_3$: skin temperature (K) | |
| $p_4$: total column water vapor (kg/kg) | |
| $p_5$: 10-1 hPa thickness | $p'_2$: $clw \times (\cos\theta)^2$ |
| $p_6$: 50-5 hPa thickness | |
| $p_7$: surface wind speed (m/s) | $p'_3$: $(\sum \Delta\tau \times \Delta T)^2$ |
| $p_8$: viewing angle | |
| $p_9$: (viewing angle)$^2$ | $p'_4$: $\sum \Delta\tau \times \Delta T$ |
| $p_{10}$: (viewing angle)$^3$ | |

Table 2 Predictors used in bias correction of different infrared hyperspectral instruments for ECMWF. The $p_x$ in the table correspond to Table 1

| Instrument | Channel | Predictors |
|---|---|---|
| AIRS | All channels | $p_1$  $p_2$  $p_5$  $p_6$ |
| CrIS | Channels below the mid-troposphere | $p_8$  $p_9$  $p_{10}$ |
| | Channels above the mid-troposphere | $p_1$  $p_2$  $p_5$  $p_6$  $p_8$  $p_9$  $p_{10}$ |
| IASI | All channels | $p_1$  $p_2$  $p_8$  $p_9$  $p_{10}$ |




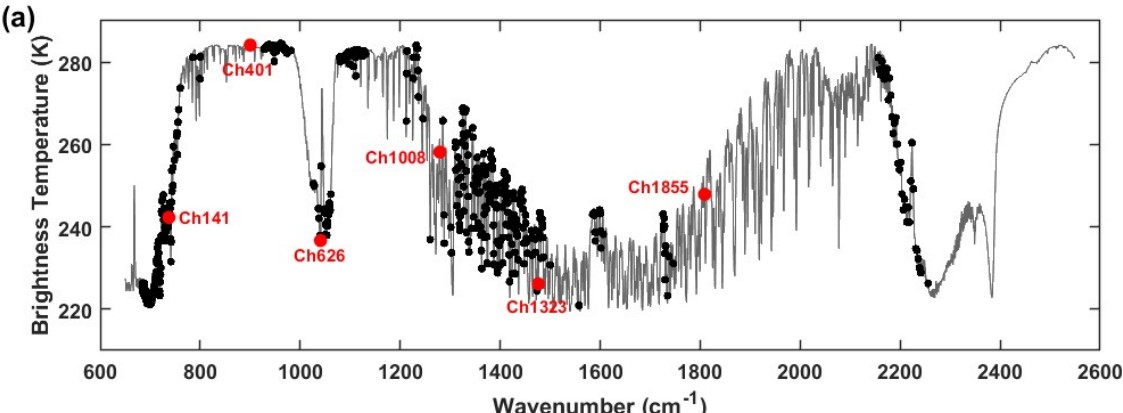


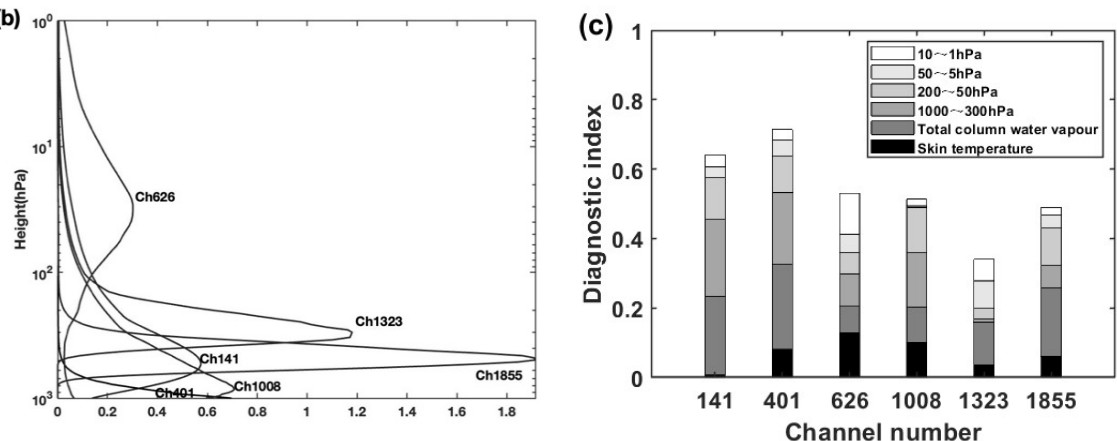


**Fig. 1**. (a) Simulated HIRAS-II brightness temperature spectrum (black curve) by the RTTOV using the US76 standard atmospheric
profile, the selected 6 typical channels (red dots) and the assimilated 485 channels (black dots). (b) weighting function of the
selected channels. (c) relevance diagnostics of bias-correction.

As we discussed earlier, model surface skin temperature, model total column water vapor, thickness of 1000-300 hPa and
thickness of 200-50 hPa are selected as predictors for bias correction. The multiple linear regression model requires a linear
relationship between continuous independent variables and dependent variables. Therefore, we examine the correlation between
HIRAS-II O-B bias and predictors $p_1$-$p_6$ based on samples from 1 to 31 January 2023. Figure 2 (a) ~ (f) show the distribution of
a typical HIRAS-II channel 1855 (1809.375cm$^{-1}$, the peak height of weight function is 500 hPa) O-B bias with predictors $p_1$-$p_6$,
respectively. In order to obtain a correct analysis of the relationship between the O-B bias and predictors $p_1$-$p_6$, the O-B biases
eliminate the influence of scan bias. The horizontal coordinate of each subgraph is the value of each predictor, the vertical
coordinate is the O-B bias and the color represents the number of samples. The black solid line in figure shows the first-order
polynomial fit of the O-B bias and predictors, the correlation coefficients are given in the subtitles of each figure. From Figure 2
(a) ~ (f), there is a certain linear correlation between the O-B bias and predictors $p_1$-$p_4$, with the highest correlation coefficient of
0.49, and there is no obvious linear relationship between the predictors $p_5$-$p_6$.



**Fig. 2** Scatterplots of (a) skin temperature, (b) total column water vapor, (c) thickness of 1000-300 hPa, (d) thickness of 200-50
hPa, (e) thickness of 50-20 hPa and (f) thickness of 10-1 hPa with respect to HIRAS-II O-B for channel 1855 (1809.375cm⁻¹, 500
hPa). Black curves show the first-order polynomial fitting. Color shade represents data number.





5. DATA ASSIMILATION SYSTEM AND EXPERIMENTAL DESIGN

This study uses WRFDA V4.4 to validate the effects of different bias correction schemes on NWP. The WRFDA model

developed by the National Center for Atmospheric Research (NCAR). It provides different methods of data assimilation and can
assimilate a wide range of observations. The WRF three-dimensional variational data assimilation system (WRF-3DVar)
minimizes the so-called variational cost function $J(x)$ as Equation (6) (Barker et al., 2004).
$$J(x) = \frac{1}{2}(x - x_b)^T B^{-1}(x - x_b) + \frac{1}{2} \times (y - H(x))^T R^{-1}(y - H(x)) \tag{6}$$

Where $x$ is the atmospheric state vectors, $x_b$ is the first guess (background), $B$ is the background error covariance, $y$ is

the observation vector, $R$ is the observation error covariance and $H$ stands for the observation operator by using RTTOV v12.3.

A new HIRAS-II data assimilation module is created in WRFDA by adding the reading and QC (The QC steps as described

in Section 3, but the MR method (Eyre and Menzel., 1989) was used for cloud detection) interfaces. A total of 485 channels are
selected for data assimilation, with specific positions in the spectrum shown as black dots in Fig. 1(a). Three parallel experiments
are designed to assess the effects of different bias correction on analysis. These experiments differed in predictors and their detailed
configurations are shown in Table 3. The initial and boundary conditions for all experiments are provided by the NECP GFS 6-h
forecast data. The simulation domain of all experiments is approximately from 0°N to 60°N and from 70°E to 150°E, with a 9 km
grid spacing on 889 × 828 horizontal grids and 60 vertical levels up to 1 hPa. The bias correction coefficients used for experiments
are fitted by the predictors and HIRAS-II O-B biases obtained from 17 to 31 July 2023, and the fitted coefficients are used in 1-
month assimilation experiments from 17 to 31 August 2023.

Table 3 The setting of three experiments

| Experiment | Description | Predictors Used in BC Scheme |
|---|---|---|
| NO BC | GFS data + conventional data + HIRAS-II data (without BC) | |
| EXP-GSI | GFS data + conventional data + HIRAS-II data (with BC) | $p'_0$: 1 <br> $p'_1$: $\frac{1}{10} \times (\frac{1}{\cos\theta} - 1)^2 - 0.015$ <br> $p'_2$: $(\sum \Delta\tau \times \Delta T)^2$ <br> $p'_3$: $\sum \Delta\tau \times \Delta T$ |
| EXP-2 | GFS data + conventional data + HIRAS-II data (with BC) | $p_0$: 1(constant) <br> $p_2$: 200-50 hPa thickness <br> $p_1$: 1000-300 hPa thickness <br> $p_3$: skin temperature <br> $p_4$: total column water vapor |


6. RESULT

The bias correction coefficients are calculated using above method based on FY-3E/HIRAS-II observations from January 1

to January 14, 2023. Subsequently, the FY-3E/HIRAS-II O-B biases in all channels from January 15 to January 31 2023 are
corrected based on the statistical coefficients. The correction results of the selected six typical channels mentioned above are
analyzed in detail.

A



*6.1 The result of scan bias correction*


The variation of mean biases with the scan angles before and after scan-angle correction for the six typical channels is
illustrated in Fig. 3. The dot-dash line represents the bias distribution before correction and the dashed and solid lines represent the
bias distribution after correction for scan angle and air-mass, respectively. The vertical axis is the mean O-B biases and the
horizontal axis is the positions of the field of regards (FORs) for each scan line (i.e., the scan angle) with the nadir between FOR14
and FOR15. It is can be seen from Fig. 3 that the biases change with the scan angles for all channels before correction. The bias
increases as the scan angle increases, especially for the lower tropospheric water vapor channel No.1008 with value up to 1.5 K
caused by the larger scan angle with respect to nadir. Additionally, there is an asymmetrical bias distribution on both sides of the
nadir in all channels, particularly for the lower height channels 141 and 1008. This is primarily caused by the non-90° inclination
angle of the polar orbit satellite, leading to inconsistent latitude of the field of view on both sides of the scan line. The phenomenon
resulted in the HIRAS-II observation being higher on one side and a lower on the other side, which in turn causes an asymmetric
distribution of O-B bias. After the scan-angle bias correction, the mentioned biases from the limb FOR relative to nadir and the
asymmetrical biases on both sides have been eliminated and the mean bias of each scan position is basically consistent. However,
there are still some residual biases (dotted line) in the properties of air mass caused by inaccurate simulation (some channels
reaching 0.5-1 K). It can be clearly seen from the black solid line in Figure 3 that the mean biases of all scan positions in all
channels approach 0 K after the air-mass correction.

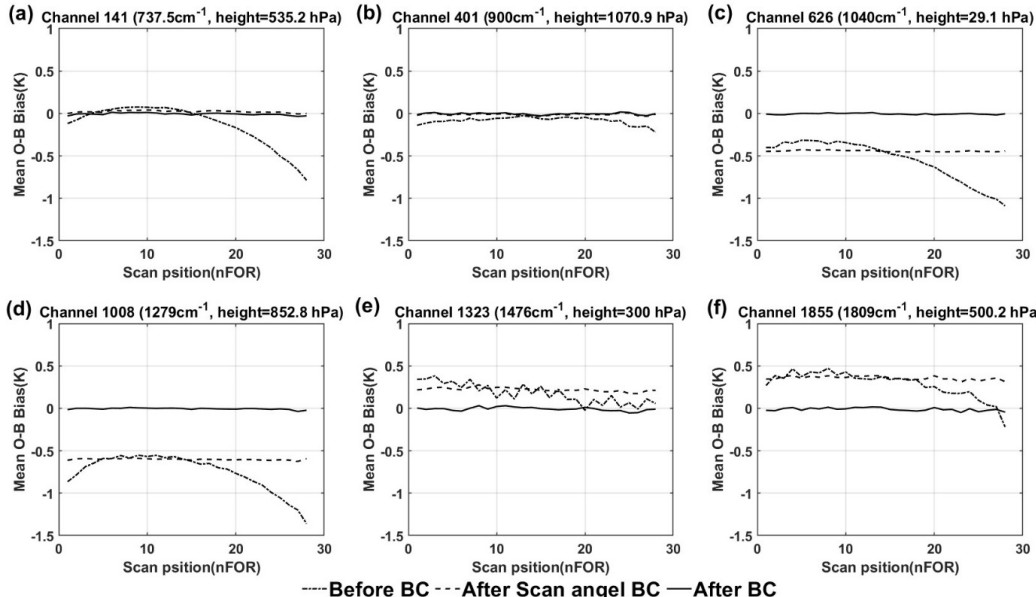


**Fig. 3** The mean O-B bias of HIRAS- II varies with the scan position before and after the bias correction. The subplots are channel
(a) No. 141, (b) 401, (c) 626, (d) 1008, (e) 1323 and (f) 1855 in turn. (The dot-dashed line represents the O-B bias before bias
correction, while the dotted line and the solid line are the results after the scan-angle and air-mass correction, respectively.)
*6.2 Comparison with GSI's air-mass bias correction scheme*
Based on the O-B biases of each HIRAS-II channel after scan-angle bias correction from 1 to 14 January 2023 and
corresponding air-mass predictors (model surface skin temperature, model total column water vapor, thickness of 1000-300 hPa
and thickness of 200-50 hPa) data, the air-mass correction coefficients are fitted using Equation (3). Then the coefficients are
applied for air-mass bias correction (referred to as EXP-2). Figures 4 (a) to (f) and (g) to (l) show the scatterplots of observed





brightness temperature (O) and background simulated brightness temperature (B) for the six representative channels before and
after the air-mass bias correction. The dashed line represents the y=x contour and the color shade represent the density of the
radiation data number. The values in each subplot are the mean value and standard deviation of O-B. From Figure 4 (a), (b) and
(d), it is evident that significant negative biases are exhibited in channels 141, 401, and 1008 (with a lower height of weight function)
when the scene temperature is high before the EXP-2 correction, while water vapor channels with a higher height of weight function
show a relatively warm bias compared to the background simulation (with mean biases ranging from 0.2 to 0.4 K). The scatters of
these channels are all concentrated and evenly distributed near the y=x contour after air-mass correction, with mean bias close to
0 K and a significantly reduced standard deviation.

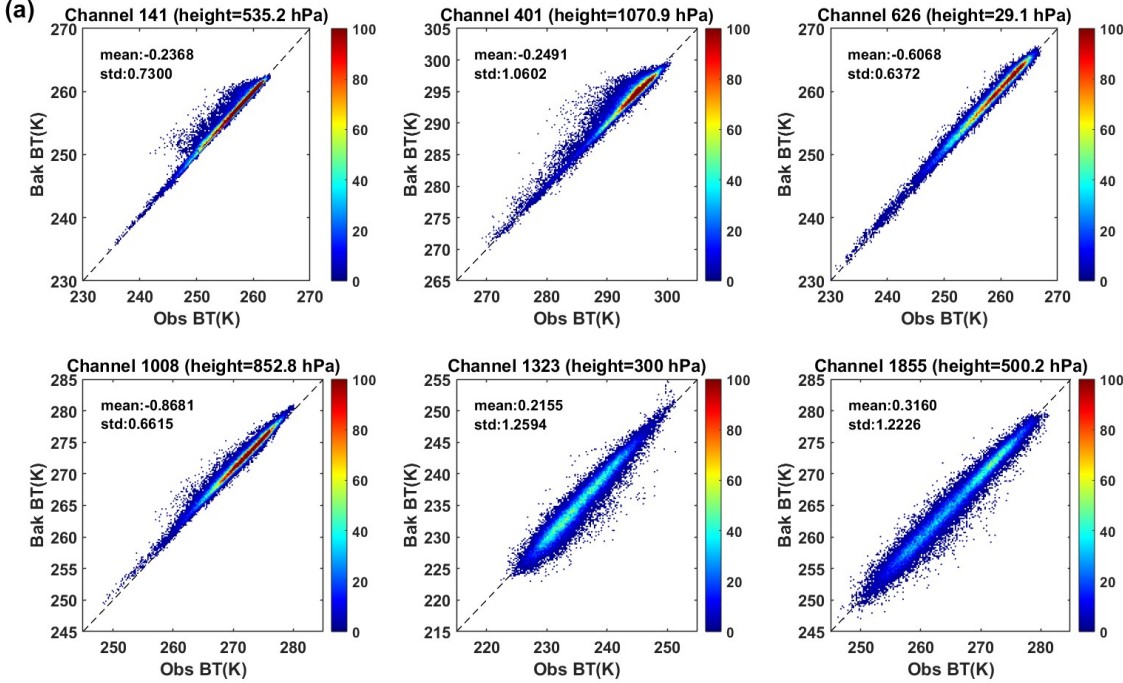

302



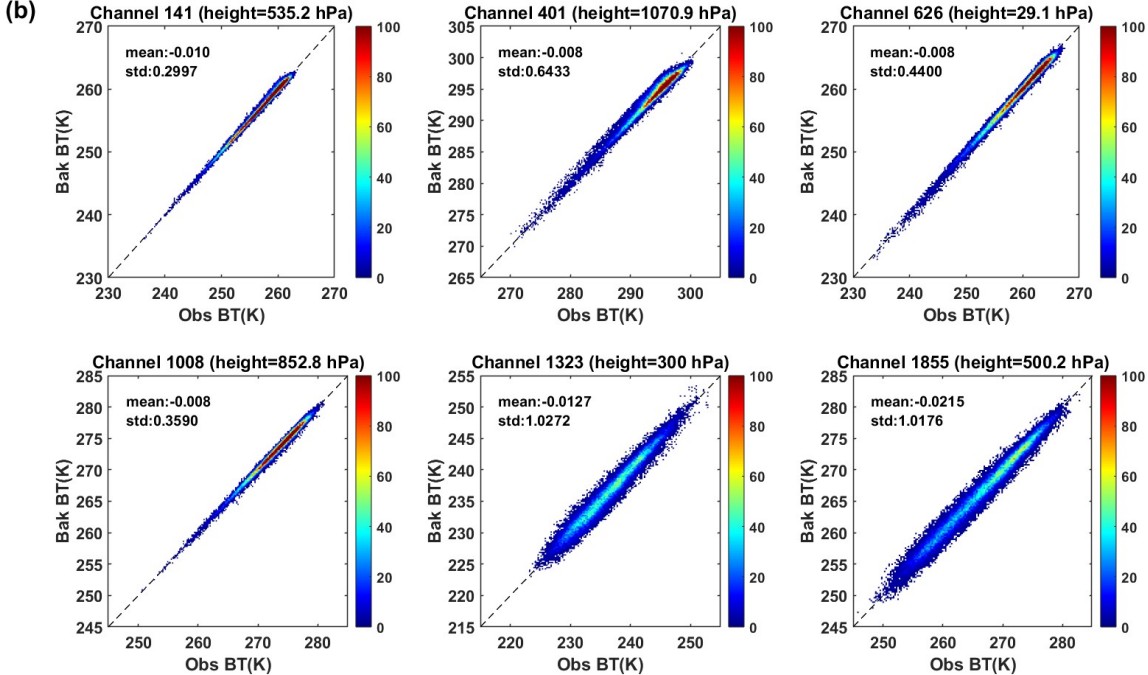

**Fig. 4** The scatterplots of observed versus RTTOV simulated brightness temperature (a) before and (b) after air-mass bias correction for HIRAS-II channel No. 141, 401, 626, 1008, 1323 and 1855. Color shade is the data number.

The bias correction scheme proposed in this paper has been compared with the air-mass bias correction scheme of NCEP-GSI to further evaluate its correction effectiveness. The air-mass predictors used in GSI for infrared instruments are $p_0'$, $p_1'$, $p_3'$, and $p_4'$ listed in Table 1 (hereafter referred to as EXP-GSI). Figure 5 illustrates the histogram of O-B biases before and after correction using two different schemes. The x-axis corresponds to the O-B bias and y-axis is the probability density function (PDF) of O-B. The black curves in the figure represents the observation residuals O-B before correction and the red and blue curves are the O-B residuals after correction using the EXP-2 and EXP-GSI schemes, respectively. The EXP-GSI scheme still displays notable positive biases in most channels after correction, especially in the channels with a height of weight function below 500 hPa. Considering that ninety percent of atmospheric moisture is confined below 500 hPa, the omission of water vapor as a predictor in the EXP-GSI scheme could potentially explain the suboptimal correction results. In contrast, the biases in all channels of the EXP-2 scheme are distributed close to a Gaussian distribution centered at zero, with the most significant correction effect in channel 1008. However, the corrected bias distribution of window channel 401 still exhibits a long tail on the left side, which may be attributed to the incomplete removal of cloud contamination scene during quality control.

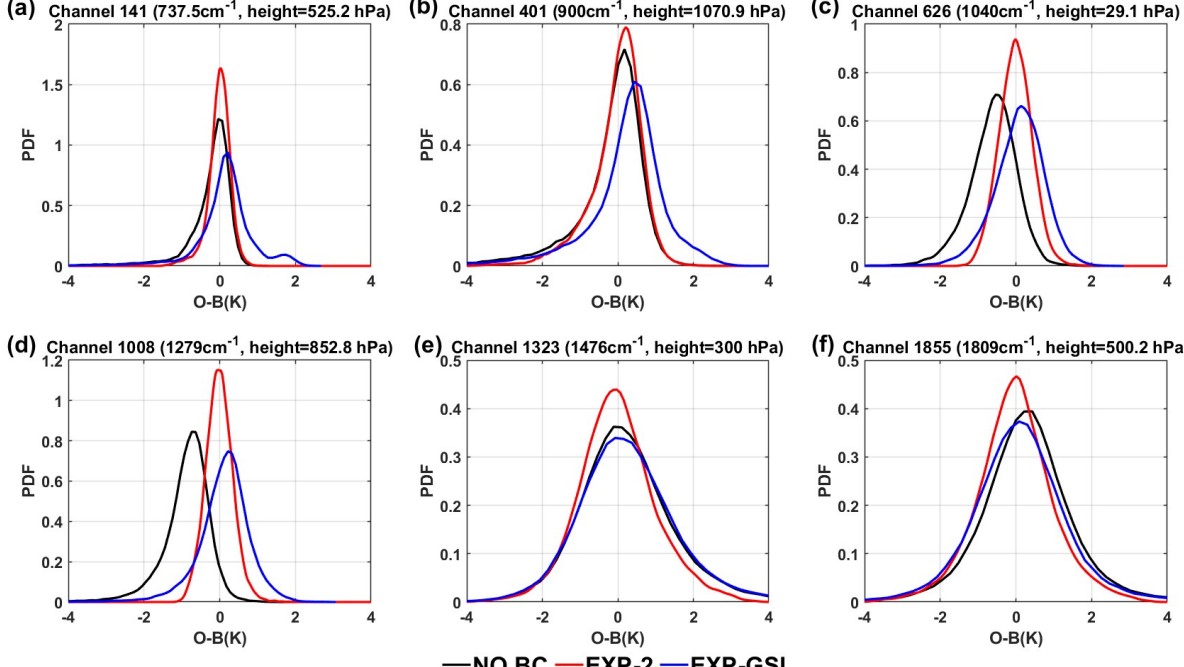

**Fig. 5** The probability density function of O-B bias before (the black curves) and after bias correction for HIRAS-2 channel (a) No. 141, (b) 401, (c) 626, (d) 1008, (e) 1323 and (f) 1855. (the red curves for EXP-2 BC and blue curves for EXP-GSI).

In order to examine whether the distribution of O-B biases after correction satisfies a normal distribution with a mean of 0, Table 4 presents the values before and after correction for these two schemes. Mean represents the mean value, STD represents the standard deviation, Kurtosis value indicates the steepness of the sample distribution and Skewness is the asymmetry of the sample distribution. Kurtosis of 3 and skewness of 0 indicate a normal distribution. It can be seen that the kurtosis and skewness values in all channels after EXP-GSI correction have not changed significantly, while both the observation residuals and standard deviations in all channels show a significant reduction after correction by EXP-2. The O-B bias of all other channels expect for channel 141 with higher kurtosis 5.4 are close to a normal distribution with kurtosis value 3 and skewness value 0. This indicates a significant improvement in the correction effect for EXP-2.

Table 4 The statistics before and after bias correction.

| Channels | Experiments | Mean | STD | Kurtosis | Skewness |
|---|---|---|---|---|---|
| | No BC | -0.6328 | 0.7300 | 23.3652 | -3.6373 |
| 141 | EXP-2 | -0.0107 | 0.2997 | 5.4265 | -0.8891 |
| | EXP-GSI | 0.1281 | 0.8424 | 22.9113 | -1.9624 |
| | NO BC | -0.2491 | 1.0602 | 16.3578 | -2.6633 |
| 401 | EXP-2 | -0.0081 | 0.6433 | 3.8705 | -0.7459 |
| | EXP-GSI | 0.2039 | 1.1362 | 17.9520 | -1.8301 |
| | NO BC | -0.6068 | 0.6372 | 5.8011 | -0.7712 |
| 626 | EXP-2 | -0.0089 | 0.4400 | 3.3943 | 0.1547 |





| | | | | | |
|---|---|---|---|---|---|
| | EXP-GSI | 0.0815 | 0.6713 | 7.0669 | -0.5527 |
| | NO BC | -0.8681 | 0.6615 | 13.0067 | -1.9956 |
| 1008 | EXP-2 | -0.0087 | 0.3590 | 3.6878 | 0.2906 |
| | EXP-GSI | 0.1052 | 0.7070 | 16.5003 | -1.2353 |
| | NO BC | 0.2155 | 1.2594 | 6.5173 | 0.4414 |
| 1323 | EXP-2 | -0.0127 | 1.0272 | 3.5891 | 0.1700 |
| | EXP-GSI | 0.2904 | 1.3913 | 7.1306 | 0.8153 |
| | NO BC | 0.3160 | 1.2226 | 7.3852 | 0.4690 |
| 1885 | EXP-2 | -0.0215 | 1.0176 | 3.7267 | -0.0030 |
| | EXP-GSI | 0.1112 | 1.3188 | 6.4132 | 0.4703 |


*6.3 Analysis results from 1-month experiments*

It is important to verify the potential impact of different BC methods on HIRAS-II radiation data assimilation. The analyzed
fields will deviate from the true state if the model first guess or observation contain systematic errors. This study assesses the effect
of different bias correction schemes on NWP based on 1-month assimilation experiments from 1 to 31 August, 2023. The mean
O-B bias and mean O-A (observation minus analysis) bias at 485 assimilated channels during the 1-month experiments period after
QC are plotted in Figure 6 (a) and (b), respectively. The horizontal coordinate is the ordinal numbers of the assimilation channels,
arranged from longwave to shortwave. The vertical coordinate is the mean bias. The black curves, blue curves and red curves in
figures represent the results of experiment NO BC, experiment EXP-GSI and experiment EXP-2, respectively. As shown in Figure
6 (a), the majority of the HIRAS channels without bias correction exhibit significant negative biases up to -2.5 K. The O3 absorption
bands from channel NO.161 to NO.182 with an average bias of 5 K due to the fixed default O3 profiles used in RTTOV. Although
the EXP-GSI scheme can reduce the absolute deviation of most channels, it still shows a certain positive deviation. It is consistent
with the results shown in Figure 5. After EXP-2 bias correction, the average biases of almost all channels are around 0 and only
channel NO.183 to NO.223 and channel NO.452 to NO.472 still have a small negative bias (maximum not exceeding -0.32 K).
This indicates that EXP-2 scheme is effective in the bias correction for HIRAS-II assimilated channels. From Fig. 6 (b), there is
obvious difference in O-A biases with and without bias correction. The mean O-A bias of the experiment without bias correction
(NO BC) shows large deviations in the O3 absorption band (channel NO.161 to NO.182), the near-surface water vapor absorption
band (channel NO.189 to NO.297) and the shortwave CO2 absorption band (channel NO.452 to NO.485). There is a significant
improvement in O-A for all channels relative to O-B after bias correction (both for EXP- GSI and EXP-2). The mean O-A of EXP-
2 (red curve) is essentially 0 for all channels except the O3 absorption band and the shortwave CO2 absorption band. It shows that
the    EXP-2    scheme    is    effective    in    improving    the    analysis.

**Fig. 6** (a) Mean O-B biases for 485 assimilation channels without BC (black curve), with the EXP-GSI BC (blue curve) and with the EXP-2 BC (red curve), (b) Mean O-A biases for 485 assimilation channels without BC (black curve), with the EXP-GSI BC (blue curve) and with the EXP-2 BC (red curve). The results are sampled from the 1-month assimilation experiments from 1 to 31 August, 2023. Only data that passed the QC process are used.

In order to specifically analyze the improvement of the analysis by different scheme, a data assimilation experiment at 0000 UTC 7 August 2023 was selected for analysis. Figure 7 shows the spatial distribution of HIRAS-II O-B biases. The colors in figures represent the values of the O-B bias, with colder colors being negative bias and warmer colors being positive bias. The shading indicates the observed FY-4A AGRI brightness temperature of the window channel 13 at a central wavelength of 12 μm. Figures 7 (a)~(c) show the distribution of O-B biases passed QC in 900 cm$^{-1}$ channel (1000 hPa) for the NOBC, EXP-GSI and EXP-2, respectively. Figures 7 (d)~(f) are as above but for 1476 cm$^{-1}$ channel (300 hPa). It can be seen that the O-B biases without BC have either cold or warm bias (Fig. 7 (a) and (d)). The most O-B biases after EXP-GSI BC are slightly warmer (Fig. 7 (b) and (e)), especially in the region from 0°N to 30°N. The overall O-B biases after EXP-2 BC is near 0 (Fig. 7 (c) and (f)) and the correction effect is a significant.



To further validate DA results, the analysis is verified against the ERA5 (ECMWF Reanalysis version 5) 0.25° × 0.25°
reanalysis data for each experiment. Figure 8 (a)~(d) give the Root Mean Squared Errors (RMSE) vertical profiles of the
temperature, specific humidity, U-wind and V-wind for the analysis and ERA5 at 0000 UTC 7 August 2023, respectively. The
dotted line represents experiment NOBC, the dashed line represents experiment EXP-GSI and the solid line represents experiment
EXP-2. The result shows that the RMSE of all variables after BC is significantly smaller than NO BC from the surface to the
troposphere. It indicates that bias correction can improve the quality of the atmospheric variables obtained from the analysis. In
the comparison of the two bias correction schemes, it can be seen that the RMSEs of all variables of EXP-2 are better than those
of EXP-GSI in the near surface (below 850 hPa) and troposphere (from 200 to 400 hPa), especially in temperature. This may be
due to the warmer O-B biases after the EXP-GSI correction.

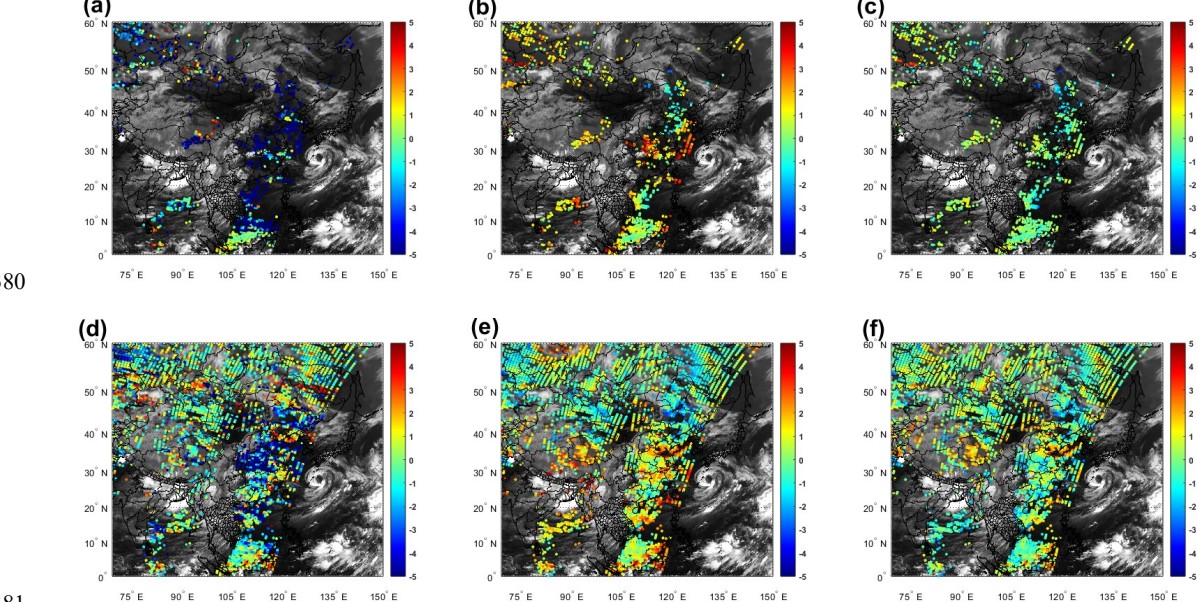

**Fig. 7** Spatial distributions of the HIRAS-II O-B biases without BC (left), with the EXP-GSI BC (middle) and with the EXP-2 BC
(right) for (a)~(c) the 900 cm$^{-1}$ channel (1000 hPa) and (d)~(f) 1476 cm$^{-1}$ channel (300 hPa) at 0000 UTC 7 August 2023. The
shading indicates the observed FY-4A AGRI brightness temperature of the window channel 13 at a central wavelength of 12 μm.





**Fig. 8** The RMSE vertical profiles of the analyzed fields from DA without BC (dotted lines), with the EXP-GSI BC (dash lines) and with the EXP-2 BC (solid lines), respectively, verified against the ERA5 for (a) temperature, (b) specific humidity, (c) U-wind and (d) V-wind valid at 0000 UTC 7 August 2023.

## 7. CONCLUSIONS

This paper establishes a two-step bias correction scheme for the innovation vector (O-B) based on the FY-3E/HIRAS-II radiations data from January 1 to January 31, 2023. Furthermore, a cross-comparison is conducted with the NCEP-GSI's air-mass bias correction scheme to objectively evaluate the effectiveness of the scheme. In addition, we briefly investigate the effectiveness of this scheme in the data assimilation system. The main conclusions are as follows:

(1) Due to the high sensitivity of the FY-3E/HIRAS-II innovation vector (O-B) to instrument scan angles, it is imperative to perform a scan bias correction. The distribution of scan biases is independent of latitude, so the division of the latitude band





is not necessary during scan correction. The biases of limb FOV with respect to nadir and the asymmetry in biases on the two
sides of the scan lines have been eliminated after scan-angle correction.
(2) The air-mass biases can be effectively eliminated by selecting the optimal combination of air-mass predictors in this
study based on correlation evaluation. The systematic biases between the observed brightness temperature and the simulated
brightness temperature in all channels are reduced and the standard deviation is also significantly decreased after correction.
Additionally, the O-B biases basically follow a Gaussian distribution with a mean of 0.
(3) The correction effect of the NCEP-GSI's air-mass bias correction scheme to these channels mid-lower tropospheric layer
channels (with a height of weight function below 500 hPa) is unsatisfactory. This could be attributed to the omission of total
column water vapor as a predictor in this scheme.
(4) Bias correction can significantly improve the quality of the analysis. The EXP-2 scheme has a significant improvement
for the analysis at upper air and near surface.
This bias correction scheme is just a preliminary experiment for the FY-3E/HIRAS-II data assimilation. At present, the off-
line static correction is adopted and the variation of O-B bias with time and weather system is not considered. The upcoming step
will involve implementing a variational bias correction scheme, the bias correction coefficient will change with time and weather
system. In addition, this study only validates the effectiveness of the scheme for HIRAS-II and will continue to validate its
applicability for similar infrared instruments in the future. Finally, this study only briefly evaluates the impact of the bias correction
scheme on data assimilation system. The impact on NWP will be further evaluated in the future in actual extreme weather individual
cases (e.g., convective precipitation and typhoons).

415                **Acknowledgment**
The authors gratefully acknowledge the financial supports by the National Natural Science Foundation of China under Grant
numbers 41975028 and Postgraduate Research & Practice Innovation Program of Jiangsu Province under Grant numbers
418    1064052301022.

419                **Author contributions**
LG planned the campaign; HC performed the measurements; HC analyzed the data; LG and HC wrote the manuscript draft;
LG reviewed and edited the manuscript.

422                **Competing interests**
The authors declare that they have no conflict of interest.

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
