# Peer review of "A Bias Correction Scheme for FY-3E/HIRAS-II Observation Data Assimilation"

_Atmospheric Measurement Techniques, 2024_

## Referee Comment (RC1)

**Review of the manuscript "A Bias Correction Scheme for FY-3E/HIRAS-II Observation Data Assimilation (Chen and Guan, 2024)"**

**General comments**

This paper focuses on a bias correction scheme for data assimilation of observation from FY-3E/HIRAS-II and proposes model surface skin temperature, model total column water vapor, thickness of 1000-300 hPa and thickness of 200-50 hPa as the optimal combination of air-mass predictors. It is especially noteworthy to compare effects of different air-mass predictors on analysis in data assimilation because there seems to be no paper to explore the optimal combination of air-mass predictors for hyperspectral infrared sounders while different NWP centers use different air-mass predictors. However, the paper needs polishing more in some respects. Model biases should not be corrected because the correction could cause loss of observation information and reinforce the systematic model errors (Auligné et al, 2007). Ideally, model biases should be handled explicitly by the system (e.g., weak-constraint 4D-Var). It is important to verify analysis of assimilation experiments by using conventional data and/or independent model analysis data to confirm that systematic model errors do not get larger. In some cases, model biases could be smaller against truth while the biases and the Root Mean Squared Errors (RMSEs) could be larger against own analysis when observation data is assimilated with bias correction. In section 6.3 of the paper, it is necessary to verify analysis those independent data (because I know from the author's comment that the analysis is verified against the NCEP FNL (Final) 0.25° × 0.25° analysis data for each experiment, not ERA5). Many descriptions in this manuscript do not follow the AMT guideline and need polishing.

**Specific comments**

L79: There is another paper (Liu et al., 2024) about data assimilation of FY-3E/HIRAS-II, and model total column water vapor is not used as air-mass predictors in that paper. It could be important to quote the paper to clarify the importance of this paper.

L109: In the manuscript, the observation data in clear-sky scene is selected to estimate bias correction and assimilate. There is no specific description that radiation is simulated assuming clear-sky condition, but I guess that clear-sky condition is assumed in the simulation of radiation. It is necessary to clarify about it in the manuscript. For example, the phrase 'assuming clear-sky condition' could be added at the end of the sentence in L109.

L117-119: The region of the training data seems to be limited by the region of FY-4A/AGRI cloud mask product. Where is the specific region when expressed by longitude and latitude?

L130-131: Why is the central FOV within each FOR selected for data thinning? Is the central FOV the best in some aspects of data quality?

L143-144: It is better to explain explicitly in section 4 in advance that the offline static correction is adopted, although section 7 includes the statement.

L174: The definitions of thickness and $t_v$ are different from those generally used in meteorology. Please check if Eq. (4) and Eq. (5) are correct.

For your information, the definitions in meteorology are shown below.

Atmospheric thickness for dry air is defined as following according to meteorological textbooks (e.g. from Eq. (1.30) in Holton et al. (2012)).

$$\text{Thickness} = \frac{R}{g} \int_{p_2}^{p_1} t\, \mathrm{d}\ln p \quad (p1 > p2)$$

The variables $R$, $g$, $p_B$, $p_U$, $t$ and $p$ are the gas constant for dry air (287.0 J K$^{-1}$ kg$^{-1}$), the gravitational acceleration (9.8 N kg$^{-1}$), pressures at the bottom and the upper of an atmospheric layer, temperature and pressure, respectively.

Assume the atmospheric model has N layers.

$$\text{Thickness} = \frac{R}{g} \sum_i^N t(i)\Delta(\ln p)_i$$

$$= \frac{R}{g} \sum_i^N t(i)\ln\frac{p_B}{p_U}$$

The variables $p_B$ and $p_U$ are pressures at the bottom and the upper of each layer, respectively. For moist air, virtual temperature $t_v$ is generally used instead of temperature.

$$\text{Thickness} = \frac{R}{g} \sum_i^N t_v(i)\ln\frac{p_B}{p_U}$$

$$t_v(i) = [1.0 + 0.608q(i)]t(i)$$

The variable is mixing ratio.

L204-206: Are the diagnostic coefficients the coefficients of determination?

Fig.1: The assimilated channels should be listed in a table in an appendix to clarify.

L253-254: In L210-211, there is the sentence 'Data assimilation systems generally do not assimilate strong $O_3$ absorption channels at present and the water vapor content in upper atmosphere is scarce, so a predictor combination including model surface skin temperature, model total column water vapor, thickness of 1000-300 hPa and thickness of 200-50 hPa is selected to correct the air-mass biases for HIRAS-II in this research.' Why are $O_3$ channels assimilated in the experiments?

L265-L330: These sentences follow section 5, 'DATA ASSIMILATION SYSTEM AND EXPERIMENTAL DESIGN'. Some people may misunderstand that these sentences explain results of data assimilation experiments. I suggest explaining these sentences in section 4 to avoid confusing.

About data availability and code availability: Authors are required to provide a statement on how their underlying software can be accessed. This must be placed as the section "Code availability" at the end of the manuscript. In the manuscript, the data corresponds to RTTOV v12.3 and WARFDA V4.4. Also, authors are required to provide a statement on how their underlying research data can be accessed. This must be placed as the section "Data availability" at the end of the manuscript. In the manuscript, the data corresponds to FY-3E/HIRAS-II Level 1 data, FY-4A/AGRI Level 2 cloud mask product, NCEP GFS forecast data and analysis data (NCEP FNL (Final) analysis data?) for verification. See the data policy page and the guideline page on the AMT website.

**Minor comments**

In the manuscript, many descriptions do not follow the AMT guideline. Please check the guideline.

L85: '... a one-month assimilation experiment.' -> "... one-month assimilation experiments.' because three assimilation experiments were run.

L89: '3041 channels' -> '3041 channels (after apodization)' to clarify the number of channels is after apodization

L98: The link has expired.

L108: A period is necessary after '(Saunders et al., 2018)'.

L164: Vectors should be printed in bold italics.

L175: The unit of the gas constant is J $K^{-1}$ $kg^{-1}$. Units must be written exponentially. 'N/kg' -> 'N $kg^{-1}$'

L196 and L201: A space must be included between number and unit in each description, '737.5$cm^{-1}$, 900$cm^{-1}$, 1040$cm^{-1}$, 1279.375$cm^{-1}$, 1476.25$cm^{-1}$ and 1809.375$cm^{-1}$'.

Fig.1 (c) and Fig.2 (b): The spelling of the word should be consistent in the manuscript. 'water vapour' -> 'water vapor'

L231 and L242: A space must be included between number and unit in the description, '1809.375cm$^{-1}$'.

Fig.2 (b): Units must be written exponentially. 'kg/kg' -> 'kg kg$^{-1}$'

L248 and L292: In the text, equations should be referred to by the abbreviation "Eq." and the respective number in parentheses. However, when the reference comes at the beginning of a sentence, the unabbreviated word "Equation" should be used. 'Equation (6)' -> 'Eq. (6)' and 'Equation (3)' -> 'Eq. (3)'

L249: Matrices should be printed in boldface, and vectors should be printed in bold italics. The multiplication sign in the second term can be omitted as well as that in first term.

L257: Coordinates need a degree sign and a space when naming the direction. The description 'from 0°N to 60°N and from 70°E to 150°E' does not follow the format.

L260: The period of assimilation experiments seems wrong. The period '17 to 31 August 2023' -> '1 to 31 August 2023'

L265-266: '… from January 1 to January 14, 2023.' and '… from January 15 to January 31 2023 …' -> '… from 1 January to14 January, 2023.' and '… from 15 January to 31 January 2023 …', respectively

Fig.3 and Fig.5: A space must be included between number and unit in the labels such as '737.5 cm$^{-1}$'.

L296-297: 'From Figure 4 (a), (b) and (d), …' But, there is no Figure 4 (d).

L298: In the sentence '…, while water vapor channels with …', are these water vapor channels channel 1323 and 1855?

Fig.4: If the figures are between 2 pages, a caption is necessary in each page. Furthermore, it may be better to use the words 'Observed BT' and 'Simulated BT' rather than 'Obs BT' and 'Bak BT' respectively in the axis labels.

Table4: If the table is between 2 pages, the table title is necessary in each page.

L342, L343, L349 and L352: 'O3' -> 'O$_3$'

L350 and L352: 'CO2'-> 'CO$_2$'

L353: The sentence should be aligned left.

L368: Coordinates need a degree sign and a space when naming the direction. The description '0°N to 30°N' does not follow the format.

L361: The sentence '..., a data assimilation experiment at 0000 UTC 7 August 2023 was...' -> '...., three data assimilation experiments at 0000 UTC 7 August 2023 were ...'?

L369: '... is a significant.' -> '... is significant.'

L362-364 and L382-384: The word colors generally include white and black. The word shading generally means not only black shading but also colorful shading. To clarify the words, 'the colors' -> 'the colored dots', 'spatial distributions' -> 'spatial distributions (colored dots)' and 'the shading' -> 'the black shading'

Fig.8: Units must be written exponentially. 'g/kg' -> 'g kg$^{-1}$' and 'm/s' -> 'm s$^{-1}$'

L391: '... from January 1 to January 31, 2023 ...' -> '... from 1 January to 31 January, 2023 ...'

[References]

Auligné, T., McNally, A.P. and Dee, D.P. (2007), Adaptive bias correction for satellite data in a numerical weather prediction system. Q.J.R. Meteorol. Soc., 133: 631-642. https://doi.org/10.1002/qj.56

Holton, J. R. and G. J. Hakim, 2012, An Introduction to Dynamic Meteorology, Fifth Edition. Academic Press, 552pp.

Liu, R., Lu, Q., Wu, C., Ni, Z., Wang, F. (2024), Assimilation of Hyperspectral Infrared Atmospheric Sounder Data of FengYun-3E Satellite and Assessment of Its Impact on Analyses and Forecasts. *Remote Sensing*, *16*(5), 908. https://doi.org/10.3390/rs16050908

Data policy page in the AMT website: https://www.atmospheric-measurement-techniques.net/policies/data_policy.html

Guideline page in the AMT website: https://www.atmospheric-measurement-techniques.net/submission.html

Thanks,

---

## Author Comment (AC2)

Response of the manuscript "A Bias Correction Scheme for FY-3E/HIRAS-II Observation Data Assimilation (Chen and Guan, 2024)"

Thank you for your professional comments. To confirm that systematic model errors do not get larger, we compare each assimilated analysis with ERA5 and calculate the average RMSE profile of temperature and water vapor in the revised manuscript. We have also made corresponding revisions to many details that do not follow the AMT guidelines. Furthermore, we will also supplement the "Code availability" and "Data availability" in the revised manuscript. Below are the specific responses, highlighted in blue:

Specific comments

1. L79: There is another paper (Liu et al., 2024) about data assimilation of FY-3E/HIRAS-II, and model total column water vapor is not used as air-mass predictors in that paper. It could be important to quote the paper to clarify the importance of this paper.

Re: We read this paper and decided to quote it in revised manuscript.

2. L109: In the manuscript, the observation data in clear-sky scene is selected to estimate bias correction and assimilate. There is no specific description that radiation is simulated assuming clear-sky condition, but I guess that clear-sky condition is assumed in the simulation of radiation. It is necessary to clarify about it in the manuscript. For example, the phrase 'assuming clear-sky condition' could be added at the end of the sentence in L109.

Re: Yes, the radiation is simulated assuming clear-sky condition. We will add the phrase 'assuming clear-sky condition' at the end of the sentence in L109 to illustrate it.

3. L117-119: The region of the training data seems to be limited by the region of FY-4A/AGRI cloud mask product. Where is the specific region when expressed by longitude and latitude?

Re: Yes, the spatial matching of FY-4A/AGRI and FY-3E/HIRAS-II has regional limitations. When the viewing angle of the satellite is large, the field of view becomes distorted and spatial resolution decreases. To minimize the impact of field-of-view deviation, this study selects only the observations from the FY-4A/AGRI with a zenith angle less than 60 ° (The region is approximately from 55 °S to 55 °N and from 50 °E to 155 °E) for matching. The specific latitude and longitude ranges are add in revised manuscript.

4. L130-131: Why is the central FOV within each FOR selected for data thinning? Is the central FOV the best in some aspects of data quality?

Re: The central FOV is chosen mainly because it is located in the center of FOR and has good observation quality. For example, figure 1 shows the NEdT (Noise Equivalent Delta Temperature) spectrum distribution of FY-3E/HIRAS-II measured at 2010 UTC 08 Dec 2021. Different colored lines are different FOVs. Figure 1 shows that the NEdT for FOV5 is minimal in almost all channels.

[Figure]

Fig 1. The NEDT distribution of FY-3E HIRAS-II on 8 December,2021.

5. L143-144: It is better to explain explicitly in section 4 in advance that the offline static correction is adopted, although section 7 includes the statement.

Re: We will explain explicitly in section 4 in advance that the offline static correction is adopted.

6. L174: The definitions of thickness and $t_v$ are different from those generally used in meteorology. Please check if Eq. (4) and Eq. (5) are correct.

Re: The equations we provide are based on the WRFDA source code. In WRFDA, the thickness is calculated assuming under the condition of moist air. In addition, the GFS data and RTTOV used in this study are based on the atmospheric level (not layer), so need 'full-level to half-level'. However, the equations given in paper are wrong, and we will change Eq. (4) and Eq. (5) as follows:

The Parameter thickness is calculated as

$$Pred_{thickness} = kth \times \sum_i^{N-1} tv(i) \times \ln\frac{P(i)}{P(i+1)} \qquad (4)$$

Here, $kth = gas_{constant}/gravity(gas_{constant} = 287.0\ J\ K^{-1}\ kg^{-1},\ gravity = 9.81\ N\ kg^{-1})$, $N$ is atmosphere levels, $P$ is atmospheric pressure, $tv$ is the parameter characterizing atmospheric temperature and humidity and calculated as

$$tv(i) = \frac{T(i)+T(i+1)}{2} \times \left[1.0 + 0.608 \times \frac{q(i)+q(i+1)}{2}\right] \quad (5)$$

Where, $T$ and $q$ represent RTM level temperatures and moistures, respectively.
The WRFDA code for computing thickness can be found in
https://www2.mmm.ucar.edu/wrf/users/wrfda/code_viewer/html_WRFDA_v4.3/index.html .
(htmlized code: radiance; radiance index: DA_PREDICTOR_RTTOV)

7. L204-206: Are the diagnostic coefficients the coefficients of determination?

Re: The diagnostic coefficients are not the coefficients of determination.

8. Fig.1: The assimilated channels should be listed in a table in an appendix to clarify

Re: We will give the assimilated channels in the attachment.

9. L253-254: In L210-211, there is the sentence 'Data assimilation systems generally do not

assimilate strong $O_3$ absorption channels at present and the water vapor content in upper atmosphere is scarce, so a predictor combination including model surface skin temperature, model total column water vapor, thickness of 1000-300 hPa and thickness of 200-50 hPa is selected to correct the air-mass biases for HIRAS-II in this research.' Why are O3 channels assimilated in the experiments?

Re: With the development of NWP mode, the top of the mode keeps increasing (ECMWF:0.01 hPa). O3 channels (the peak height of the weight function ranges from 5 to 20hPa) can provide upper-air meteorological information that cannot be provided by conventional data. Therefore, we still choose these channels for assimilation although the data assimilation system generally does not assimilate O3 channels at present.

10. L265-L330: These sentences follow section 5, 'DATA ASSIMILATION SYSTEM AND EXPERIMENTAL DESIGN'. Some people may misunderstand that these sentences explain results of data assimilation experiments. I suggest explaining these sentences in section 4 to avoid confusing.

Re: I agree with your comment, it will be modified in revised manuscript.

11. About data availability and code availability: Authors are required to provide a statement on how their underlying software can be accessed. This must be placed as the section "Code availability" at the end of the manuscript. In the manuscript, the data corresponds to RTTOV v12.3 and WARFDA V4.4. Also, authors are required to provide a statement on how their underlying research data can be accessed. This must be placed as the section "Data availability" at the end of the manuscript. In the manuscript, the data corresponds to FY-3E/HIRAS-II Level 1 data, FY-4A/AGRI Level 2 cloud mask product, NCEP GFS forecast data and analysis data (NCEP FNL (Final) analysis data?) for verification. See the data policy page and the guideline page on the AMT website.

Re: We will provide "Code availability" and "Data availability" at the end of the revised manuscript.

Minor comments

1. L85: '… a one-month assimilation experiment.' -> "… one-month assimilation experiments.' because three assimilation experiments were run.

Re: I agree with your comment, it will be modified in revised manuscript.

2. L89: '3041 channels'-> '3041 channels (after apodization)'to clarify the number of channels is after apodization

Re: I agree with your comment, it will be modified in revised manuscript.

3. L98: The link has expired.

Re: We update the link (https://rda.ucar.edu/datasets/d084001/).

4. L108: A period is necessary after '(Saunders et al., 2018)'.

Re: Thank you for your careful comment, it will be modified in revised manuscript.

5. L164: Vectors should be printed in bold italics.

Re: Thank you for your careful comment, it will be modified in revised manuscript.

6. L175: The unit of the gas constant is J K$^{-1}$ kg$^{-1}$. Units must be written exponentially. 'N/kg'-> 'N kg$^{-1}$'

Re: I agree with your comment, it will be modified in revised manuscript.

7. L196 and L201: A space must be included between number and unit in eachdescription, '737.5cm$^{-1}$, 900cm$^{-1}$, 1040cm$^{-1}$, 1279.375cm$^{-1}$, 1476.25cm$^{-1}$ and 1809.375cm$^{-1}$'.

Re: Thank you for your careful comment, it will be modified in revised manuscript.

8. Fig.1 (c) and Fig.2 (b): The spelling of the word should be consistent in the manuscript.'water vapour'-> 'water vapor'

Re: Thank you for your careful comment, Fig.1 (c) and Fig.2 (b) will be updated in revised manuscript.

9. L231 and L242: A space must be included between number and unit in the description, '1809.375cm-1'.

Re: Thank you for your careful comment, it will be modified in revised manuscript.

10. Fig.2 (b): Units must be written exponentially. 'kg/kg'-> 'kg kg$^{-1}$'

Re: I agree with your comment, Fig.2 (b) will be update in revised manuscript.

11. L248 and L292: In the text, equations should be referred to by the abbreviation "Eq." and the respective number in parentheses. However, when the reference comes at the beginning of a sentence, the unabbreviated word "Equation" should be used. 'Equation (6)' -> 'Eq. (6)' and 'Equation (3)' -> 'Eq. (3)'

Re: I agree with your comment, it will be modified in revised manuscript.

12. L249: Matrices should be printed in boldface, and vectors should be printed in bold italics. The multiplication sign in the second term can be omitted as well as that in first term.

Re: Thank you for your careful comment, it will be modified in revised manuscript.

13. L257: Coordinates need a degree sign and a space when naming the direction. The description 'from 0°N to 60°N and from 70°E to 150°E' does not follow the format.

Re: Thank you for your careful comment, it will be modified in revised manuscript.

14. L260: The period of assimilation experiments seems wrong. The period '17 to 31 August 2023'-> '1 to 31 August 2023'

Re: I agree with your comment, it will be modified in revised manuscript.

15. L265-266: '… from January 1 to January 14, 2023.' and '… from January 15 to January 31 2023 …'-> '… from 1 January to14 January, 2023.' and '… from 15 January to 31 January 2023 …', respectively

Re: I agree with your comment, it will be modified in revised manuscript.

16. Fig.3 and Fig.5: A space must be included between number and unit in the labels such as

'737.5 cm$^{-1}$'.

Re: Thank you for your careful comment, we update Fig.3 and Fig.5 in revised manuscript.

17. L296-297: 'From Figure 4 (a), (b) and (d), …' But, there is no Figure 4 (d).

Re: We update the Figure 4 to address the issue, each subgraph in the updated figure has its own serial number

18. L298: In the sentence '…, while water vapor channels with …', are these water vapor channels channel 1323 and 1855?

Re: Yes, we added '(Figure 4 (e-f))' after the sentence 'water vapor channels with a higher height of weight function ' to make our expression more clear.

19. Fig.4: If the figures are between 2 pages, a caption is necessary in each page. Furthermore, it may be better to use the words 'Observed BT' and 'Simulated BT'rather than 'Obs BT' and 'Bak BT'respectively in the axis labels.

Re: To address this issue, we merge the two figures into a single composite figure. In addition, we use the words 'Observed BT' and 'Simulated BT' in the axis labels, respectively.

20. Table4: If the table is between 2 pages, the table title is necessary in each page.

Re: I agree with your comment, it will be modified in revised manuscript.

21. L342, L343, L349 and L352: 'O3' -> 'O$_3$'

Re: Thank you for your careful comment, it will be modified in revised manuscript.

22. L350 and L352: 'CO2'-> 'CO$_2$'

Re: Thank you for your careful comment, it will be modified in revised manuscript.

23. L353: The sentence should be aligned left.

Re: Thank you for your careful comment, it will be modified in revised manuscript.

24. L368: Coordinates need a degree sign and a space when naming the direction. The description '0°N to 30°N' does not follow the format.

Re: Thank you for your careful comment, it will be modified in revised manuscript.

25. L361: The sentence '…, a data assimilation experiment at 0000 UTC 7 August 2023 was…'-> '…., three data assimilation experiments at 0000 UTC 7 August 2023 were …'?

Re: Yes, it will be modified in revised manuscript.

26. L369: '… is a significant.'-> '… is significant.'

Re: It will be modified in revised manuscript.

27. L362-364 and L382-384: The word colors generally include white and black. The word shading generally means not only black shading but also colorful shading. To clarify the words, 'the colors'-> 'the colored dots', 'spatial distributions'-> 'spatial distributions (colored dots)' and 'the shading'-> 'the black shading

Re: I agree with your comment, it will be modified in revised manuscript.

28. Fig.8: Units must be written exponentially. 'g/kg'-> 'g kg-1' and 'm/s'-> 'm s-1'

Re: I agree with your comment, it will be modified in revised manuscript.

29. L391: '… from January 1 to January 31, 2023 …'-> '… from 1 January to 31 January, 2023…

Re: It will be modified in revised manuscript.

Thank you again for your professional and meticulous comments.

---

## Author Response (AR1)

**Final author reply to the editor**

**Dear Editor:**

Thank you to the two reviewers for your professional comments, we have made detailed revisions to the manuscript based on your suggestions to meet the high quality standards of AMT. The revisions based on reviewer 1' comments are marked in blue in the revised manuscript, the revisions based on reviewer 2' comments are marked in red. The following are the reviewers' comments, the authors' responses (marked in blue) and author's changes in manuscript (marked in red).

**Comments from referee 1:**

1. Ideally, model biases should be handled explicitly by the system (e.g., weak-constraint 4D-Var). It is important to verify analysis of assimilation experiments by using conventional data and/or independent model analysis data to confirm that systematic model errors do not get larger. In some cases, model biases could be smaller against truth while the biases and the Root Mean Squared Errors (RMSEs) could be larger against own analysis when observation data is assimilated with bias correction. In section 6.3 of the paper, it is necessary to verify analysis those independent data.

Re: We compare each assimilated analysis with ERA5 and calculate the average RMSE profile of temperature and water vapor in the revised manuscript (Line 397-409).

2. L79: There is another paper about data assimilation of FY-3E/HIRAS-II, and model total column water vapor is not used as air-mass predictors in that paper. It could be important to quote the paper to clarify the importance of this paper.

Re: We read this paper and decided to quote it in Line 85-87 and Line 90-92.

3. L109: In the manuscript, the observation data in clear-sky scene is selected to estimate bias correction and assimilate. There is no specific description that radiation is simulated assuming clear-sky condition, but I guess that clear-sky condition is assumed in the simulation of radiation. It is necessary to clarify about it in the manuscript. For example, the phrase 'assuming clear-sky condition' could be added at the end of the sentence in L109.

Re: Yes, the radiation is simulated assuming clear-sky condition. We will add the phrase 'assuming clear-sky condition' at the end of the sentence in Line 95-96 to illustrate it.

4. L117-119: The region of the training data seems to be limited by the region of FY-4A/AGRI cloud mask product. Where is the specific region when expressed by longitude and latitude?

Re: Yes, the spatial matching of FY-4A/AGRI and FY-3E/HIRAS-II has regional limitations. When the viewing angle of the satellite is large, the field of view becomes distorted and spatial resolution decreases. To minimize the impact of field-of-view deviation, this study selects only the observations from the FY-4A/AGRI with a zenith angle less than 60 ° (The region is approximately

from 55 °S to 55 °N and from 50 °E to 155 °E) for matching. The specific latitude and longitude ranges are added in Line 131-134.

5. L130-131: Why is the central FOV within each FOR selected for data thinning? Is the central FOV the best in some aspects of data quality?

Re: The central FOV is chosen mainly because it is located in the center of FOR and has good observation quality. For example, figure 1 shows the NEdT (Noise Equivalent Delta Temperature) spectrum distribution of FY-3E/HIRAS-II measured at 2010 UTC 08 Dec 2021. Different colored lines are different FOVs. Figure 1 shows that the NEdT for FOV5 is minimal in almost all channels.

Fig 1. The NEDT distribution of FY-3E HIRAS-II on 8 December 2021.

6. L143-144: It is better to explain explicitly in section 4 in advance that the offline static correction is adopted, although section 7 includes the statement.

Re: We will explain explicitly in section 4 in advance (Line 158) that the offline static correction is adopted.

7. L174: The definitions of thickness and  $t_v$  are different from those generally used in meteorology. Please check if Eq. (4) and Eq. (5) are correct.

Re: The equations we provide are based on the WRFDA source code. In WRFDA, the thickness is calculated assuming under the condition of moist air. In addition, the GFS data and RTTOV used in this study are based on the atmospheric level (not layer), so need 'full-level to half-level'. However, the equations given in paper are wrong, and we will change Eq. (4) and Eq. (5) as follows (Line 196-199):

The Parameter thickness is calculated as

$$Pred_{thickness} = kth \times \sum_{i}^{N-1} tv(i) \times \ln \frac{P(i)}{P(i+1)}$$
 (4)

Here,  $kth = gas_{constant}/gravity(gas_{constant} = 287.0 J K^{-1} kg^{-1}, gravity =$

 $9.81 \, N \, kg^{-1}$ ), N is atmosphere levels, P is atmospheric pressure, tv is the parameter characterizing atmospheric temperature and humidity and calculated as

$$tv(i) = \frac{T(i) + T(i+1)}{2} \times \left[1.0 + 0.608 \times \frac{q(i) + q(i+1)}{2}\right] (5)$$

Where, *T* and *q* represent RTM level temperatures and moistures, respectively. The WRFDA code for computing thickness can be found in <a href="https://www2.mmm.ucar.edu/wrf/users/wrfda/code\_viewer/html\_wRFDA\_v4.3/index.html">https://www2.mmm.ucar.edu/wrf/users/wrfda/code\_viewer/html\_wRFDA\_v4.3/index.html</a> (htmlized code: radiance; radiance index: DA\_PREDICTOR\_RTTOV)

8. L204-206: Are the diagnostic coefficients the coefficients of determination? Re: The diagnostic coefficients are not the coefficients of determination.

9. Fig.1: The assimilated channels should be listed in a table in an appendix to clarify Re: We will give the assimilated channels in the attachment.

10. L253-254: In L210-211, there is the sentence 'Data assimilation systems generally do not assimilate strong O3 absorption channels at present and the water vapor content in upper atmosphere is scarce, so a predictor combination including model surface skin temperature, model total column water vapor, thickness of 1000-300 hPa and thickness of 200-50 hPa is selected to correct the airmass biases for HIRAS-II in this research.' Why are O3 channels assimilated in the experiments? Re: With the development of NWP mode, the top of the mode keeps increasing (ECMWF:0.01 hPa). O3 channels (the peak height of the weight function ranges from 5 to 20hPa) can provide upper-air meteorological information that cannot be provided by conventional data. Therefore, we still choose these channels for assimilation although the data assimilation system generally does not assimilate O3 channels at present.

11. L265-L330: These sentences follow section 5, 'DATA ASSIMILATION SYSTEM AND EXPERIMENTAL DESIGN'. Some people may misunderstand that these sentences explain results of data assimilation experiments. I suggest explaining these sentences in section 4 to avoid confusing.

Re: I agree with your comment, we move this part to the Line 337-354.

12. About data availability and code availability: Authors are required to provide a statement on how their underlying software can be accessed. This must be placed as the section "Code availability" at the end of the manuscript. In the manuscript, the data corresponds to RTTOV v12.3 and WARFDA V4.4. Also, authors are required to provide a statement on how their underlying research data can be accessed. This must be placed as the section "Data availability" at the end of the manuscript. In the manuscript, the data corresponds to FY-3E/HIRAS-II Level 1 data, FY-

4A/AGRI Level 2 cloud mask product, NCEP GFS forecast data and analysis data (NCEP FNL (Final) analysis data?) for verification. See the data policy page and the guideline page on the AMT website.

Re: We provide "Code availability" and "Data availability" at the end of the revised manuscript (Line 437-448).

12. L85: '... a one-month assimilation experiment.' -> "... one-month assimilation experiments.' because three assimilation experiments were run.

Re: I agree with your comment, it is modified in revised manuscript (Line 95).

13. L89: '3041 channels'-> '3041 channels (after apodization)'to clarify the number of channels is after apodization

Re: I agree with your comment, it is modified in revised manuscript (Line 100).

14. L98: The link has expired.

Re: We update the link (https://rda.ucar.edu/datasets/d084001/) (Line 109).

15. L108: A period is necessary after '(Saunders et al., 2018)'.

Re: Thank you for your careful comment, it is modified in revised manuscript (Line 119).

16. L164: Vectors should be printed in bold italics.

Re: Thank you for your careful comment, it is modified in revised manuscript (Line 184).

17. L175: The unit of the gas constant is J  $K^{-1}$   $kg^{-1}$ . Units must be written exponentially. 'N/ $kg^{-1}$ ' 'N  $kg^{-1}$ '

Re: I agree with your comment, it is modified in revised manuscript (Line 197).

18. L196 and L201: A space must be included between number and unit in each description, '737.5cm-1, 900cm-1, 1040cm-1, 1279.375cm-1, 1476.25cm-1 and 1809.375cm-1'.

Re: Thank you for your careful comment, it is modified in revised manuscript (Line 211-224).

19. Fig.1 (c) and Fig.2 (b): The spelling of the word should be consistent in the manuscript. 'water vapour'-> 'water vapor'

Re: Thank you for your careful comment, Fig.1 (c) and Fig.2 (b) are updated in revised manuscript.

20. L231 and L242: A space must be included between number and unit in the description, '1809.375cm-1'.

Re: Thank you for your careful comment, it is modified in revised manuscript (Line 253).

21. Fig.2 (b): Units must be written exponentially. 'kg/kg'-> 'kg kg-1'

Re: I agree with your comment, Fig.2 (b) is update in revised manuscript.

22. L248 and L292: In the text, equations should be referred to by the abbreviation "Eq." and the respective number in parentheses. However, when the reference comes at the beginning of a sentence, the unabbreviated word "Equation" should be used. 'Equation (6)' -> 'Eq. (6)' and 'Equation (3)' -> 'Eq. (3)'

Re: I agree with your comment, it is modified in revised manuscript (Line 342).

23. L249: Matrices should be printed in boldface, and vectors should be printed in bold italics. The multiplication sign in the second term can be omitted as well as that in first term.

Re: Thank you for your careful comment, it is modified in revised manuscript (Line 343).

24. L257: Coordinates need a degree sign and a space when naming the direction. The description 'from 0°N to 60°N and from 70°E to 150°E' does not follow the format.

Re: Thank you for your careful comment, it is modified in revised manuscript (Line 351).

25. L260: The period of assimilation experiments seems wrong. The period '17 to 31 August 2023'-> '1 to 31 August 2023'

Re: It is modified in revised manuscript (Line 354).

26. L265-266: '... from January 1 to January 14, 2023.' and '... from January 15 to January 31 2023 ...'-> '... from 1 January to 14 January, 2023.' and '... from 15 January to 31 January 2023 ...', respectively

Re: It is modified in revised manuscript (Line 267-268).

27. Fig.3 and Fig.5: A space must be included between number and unit in the labels such as '737.5 cm-1'.

Re: Thank you for your careful comment, we update Fig.3 and Fig.5 in revised manuscript.

28. L296-297: 'From Figure 4 (a), (b) and (d), ...' But, there is no Figure 4 (d).

Re: We update the Figure 4 to address the issue, each subgraph in the updated figure has its own serial number (see Figure 4).

29. L298: In the sentence '..., while water vapor channels with ...', are these water vapor channels

channel 1323 and 1855?

Re: Yes, we added '(Figure 4 (e-f))' (Line 300-301) after the sentence 'water vapor channels with a higher height of weight function 'to make our expression more clear.

30. Fig.4: If the figures are between 2 pages, a caption is necessary in each page. Furthermore, it may be better to use the words 'Observed BT' and 'Simulated BT'rather than 'Obs BT' and 'Bak BT'respectively in the axis labels.

Re: To address this issue, we merge the two figures into a single composite figure. In addition, we use the words 'Observed BT' and 'Simulated BT' in the axis labels, respectively (see Figure 4).

31. Table4: If the table is between 2 pages, the table title is necessary in each page.

Re: It is modified in revised manuscript (Line 267-268).

32. L342, L343, L349 and L352: 'O3' -> 'O3'

Re: Thank you for your careful comment, it is modified in revised manuscript (Line 364, Line 365, Line 370, Line 373).

33. L350 and L352: 'CO2'-> 'CO2'

Re: Thank you for your careful comment, it is modified in revised manuscript (Line 371, Line 374).

34. L353: The sentence should be aligned left.

Re: It is modified in revised manuscript (Line 374).

35. L368: Coordinates need a degree sign and a space when naming the direction. The description '0°N to 30°N' does not follow the format.

Re: It is modified in revised manuscript (Line 389).

36. L361: The sentence '..., a data assimilation experiment at 0000 UTC 7 August 2023 was...'-> '..., three data assimilation experiments at 0000 UTC 7 August 2023 were ...'?

Re: Yes, it is modified in revised manuscript (Line 382-383).

37. L369: '... is a significant.'-> '... is significant.'

Re: It is modified in revised manuscript (Line 390).

38. L362-364 and L382-384: The word colors generally include white and black. The word shading generally means not only black shading but also colorful shading. To clarify the words, 'the colors'-> 'the colored dots', 'spatial distributions'-> 'spatial distributions (colored dots)' and 'the shading'->

Re: I agree with your comment, it is modified in revised manuscript (Line 393).

39. Fig.8: Units must be written exponentially. 'g/kg'-> 'g kg-1' and 'm/s'-> 'm s-1' Re: It is modified in revised manuscript (see Figure 8).

40. L391: '... from January 1 to January 31, 2023 ...'-> '... from 1 January to 31 January, 2023... Re: It is modified in revised manuscript (Line 412).

**Comments from referee 2:**

**1. Scan-angle bias origin unclear.**

The paper shows a clear scan-angle bias in HIRAS-II data but does not clarify whether this originates from the satellite sensor, the radiative transfer model (e.g., RRTOV), or atmospheric mismatches between observations and simulations. Given that TOA radiances naturally depend on viewing angle (due to path length, surface emission angle-dependence, and atmospheric heterogeneity), the authors should systematically analyze and discuss these potential causes.

Re: Just as mentioned in the Line 101-104, HIRAS-II is a cross-track scanning instrument. The distribution of FY-3E/ HIRAS-II FOVs in a scan line is shown in Figure 2. The scan angle of each FOV gradually increases as the instrument scans to both sides (FOR1 and FOR28), inevitably causing a scan angle bias relative to the nadir.

"The field of view is susceptible to deformation as the scan angle increases when the satellite sensor performs a cross-track scanning to both sides of scan lines, which leads to unavoidable observation biases relative to the nadir FOV. The optical pathlength is longer as the scan angle increases, also leading to scan angle biases. In addition, although the fast radiative transfer models are become increasingly sophisticated and accurate, they can not only simulate the brightness temperature at the nadir, but also simulate the brightness temperature at other scan angles. However, atmospheric inhomogeneity increases with the increase of scan angle. If the model cannot fully simulate atmospheric inhomogeneity, it may also lead to bias in the simulated brightness temperature that depend on the scan angle." has been add to Line 161-167 to systematically analyze and discuss scan-angle bias origin.

Figure.2. The distribution of FY-3E/ HIRAS-II field of views in a scan line.

**2. Air-mass bias concept not well defined.**

The introduction should better distinguish scan-angle bias (instrumental/geometric) from airmass bias (systematic model error in representing specific air mass types). The use of predictors to correct O–B needs a clearer explanation: what physical aspects are being corrected—surface temperature, tropospheric composition, or something else?

Re: The "air-mass" is a professional term inherited from the past, referring to the collection of atmospheric state variables (e.g., temperature, water vapor, or pressure) related to O-B bias within the field of view.

"For a long time, satellite radiance data assimilation has mainly used a bias correction scheme that relies on "air-mass", which use quantities related to "air state" as predictors to implement bias correction. Its theoretical basis is that there is a linear correlation between the spatial and temporal variations of the satellite observation biases and the predictors, and the higher the correlation, the more obvious the correction effect is" is added to Line 60-63 to define the air-mass bias correction.

Line 66-68 "It is worth......bias correction" is modified to "It is worth mentioning that Harris and Kelly (2001) proposed a revolutionary bias correction scheme that divides the bias into two parts: the unavoidable scan angle bias of cross-track scanning instruments and the air-mass bias caused by the NWP mode and the fast radiative transfer mode" to distinguish the scan angle bias and air-mass bias.

"The predictor  $p_1$ ,  $p_2$ ,  $p_5$  and  $p_6$  reflect the mode background-errors at different layers and various dependencies within the forward model. The predictor  $p_3$  represents the systematic error of near-surface channels. In addition, it can compensate somewhat for the different emissivity characteristics of different surface types for these channels (Harris and Kelly, 2001). The water vapor is one of the main input components in the fast radiative transfer mode, the predictor  $p_4$  can reflect the model simulated error to some extent" is added to Line 190-194 to explain the physical aspects corrected by each predictor.

**3. Inconsistent description of data assimilation setup.**

The abstract mentions NCEP-GSI, but the experiments use WRFDA v4.4. This inconsistency must be resolved. Also, the timeline of the assimilation period is contradictory (17–31 August vs. 1–31 August 2023). This needs clarification to maintain credibility.

Re: We used the airmass predictors of NECP-GSI as the control scheme, but the assimilation experiments are carried out in the WRFDA assimilation system. This is because the observation operator of GSI assimilation system is CRTM. Since the CRTM does not provide the coefficient files of Fengyun series satellites, we cannot obtain the coefficient file of HIRAS. The observation operators of WRFDA include CRTM and RTTOV. Therefore, we choose WRFDA to conduct the assimilation experiments. The error in the assimilation time has been corrected (Line 354).

**4. Lack of physical explanation for air-mass correction.**

The correlation between the chosen predictors and O-B is shown, but there is no justification for why these predictors lead to better TOA radiance simulation. The study would benefit from showing how the air-mass correction improves key state variables (e.g., surface T, pressure, or humidity) and thereby reduces bias.

Re: Bias correction is not to better simulate radiance (B), but to meet the assumption that both observation and background errors are unbiased as required by the variational assimilation method.

As mentioned in the Line 101-104, observation and background errors can be calculated by O-B. Therefore, the ultimate goal of bias correction is to make O-B distribution follows a Gaussian normal distribution with a mean of 0 (Figure 5). Independent ERA5 (ECMWF Reanalysis version 5) objective analysis fields are selected as the true value to further evaluate the impact of different FY-3E/HIRAS-II BC schemes on the data assimilation analyses, the analysis fields with different BC schemes are compared with ERA5 in the Figure 8. It can be seen from the figure that bias correction has a improvement for the temperature and humidity analysis (closed to ERA5).

- 5. Title: FY-3E/HIRAS-II (no space after /)
  - Re: The revision has been completed (see the title).
- **6.** L26: "satellite radiations" --> satellite radiance.
  - Re: The revision has been completed (Line 29).
- 7. L34-35: "The satellite launched ...time window". This sentence is unclear. FY-3E has an early morning overpass, so how would this fill a time gap within a 6-hour assimilation window? IASI also has a morning overpass.

Re: "the time gap of satellite observations" has been modified to "the data gap of satellite observations" and "ensuring 100% coverage..." has been modified to "ensuring almost 100% coverage..." (Line 37-38). The equator crossing time (ETC) in local standard time of METOP-B/C is 0930, while that of FY-3E is 0530. Figure 3 shows that the data coverage of FY-3C (afternoon orbit) and FY-3F (morning orbit, similar to METOP-B/C) at different assimilation times. It can be seen from Figure 1 that there are data gaps at each assimilation time. Figure 4 shows the spatial coverage of FY-3C, FY-3F and FY-3E (early-morning orbit). It can be seen from Figure 4 that the three-orbit constellation can comprehensively increase the spatial coverage of polar-orbiting satellites, ensuring almost 100% coverage of satellite observations within the assimilation time window.

Fig. 3. The spatial coverage of FY-3C (red color) and FY-3F (green color) at assimilation time (a) 0000UTC, (b) 0600UTC, (c) 1200UTC and (d) 1800 UTC.

Fig. 4. The spatial coverage of FY-3C (red color), FY-3F (green color) and FY-3E (blue color) at assimilation time (a) 0000UTC, (b) 0600UTC, (c) 1200UTC and (d) 1800 UTC.

8. L46: "destroying the global NWP system" ... this is oddly phrased. The system will surely not

be destroyed, but the model state may be negatively influenced when biases persist in satellite observations. This should be rephrased.

Re: "destroying the global NWP system" has been modified to "affect the forecast accuracy of NWP" (Line 49).

9. L108: Period missing after Saunders-reference.

Re: The revision has been completed (Line 119).

10. L135: "exceeds"--> exceeding

Re: The revision has been completed (Line 150).

11. L148: rephrase to 'For each channel, the global ...position is calculated as:'

Re: The revision has been completed (Line 168).

12. L300: "The scatters" --> The scatter

Re: The revision has been completed (Line 302).

**13.** L380: the colour bar in Figure 7 is unfortunate with green hue indicating no bias. Suggest to use a cold-white-warm colour scheme, where white corresponds to zero bias.

Re: The revision has been completed (see Figure 7).

Thank you again for your professional and meticulous comments.

---

## Author Response (AR2)

**Response of the manuscript "A Bias Correction Scheme for FY-3E/HIRAS-II Observation Data Assimilation (Chen and Guan, 2024)"**

Thank you for your professional comments. We have also made corresponding revisions based on your professional comments. Below are the specific responses, highlighted in blue:

**General comments**

This paper shows verifications of two experiments with bias correction relative to the experiment without bias correction. It is important to show verifications against an experiment without assimilation of HIRAS-II to confirm the impact of the bias correction. If these verifications show degradation, additional discussions about the causes will be necessary, and it may suggest that some bias remain in the observation data.

Re: The experiment without assimilation of HIRAS-II (EXP-CONV) is added in revised manuscript. Its analysis field is also verified against the ERA5  $0.25^{\circ} \times 0.25^{\circ}$  data. The vertical RMSE (Root Mean Square Error) profiles of temperature and water vapor from EXP-CONV are added in Figure 8 (a) and (c), respectively.

Compared with experiment EXP-COVN, experiments EXP-2, EXP-GSI and EXP-NOBC assimilated a large amount of HIRAS-II data covering from the ground to the upper atmosphere. Therefore, it can be seen from Figure 8 (a) that the temperature analysis field accuracy is effectively improved (closest to ERA5), including experiment EXP-NOBC. The RMSE of temperature analysis fields from NOBC, EXP-2 and EXP-GSI significantly decrease relative to EXP-CONV in all atmosphere (Figure 8 (a)), the RMSE of water vapor analysis fields slightly decrease relative to EXP-CONV below 800 hPa (Figure 8 (c)). This result to some extent proves that HIRAS-II data have the value for assimilation.

To validate the influence of different bias correction schemes, the RMSE profiles from EXP-2 and EXP-GSI normalized by the RMSE from the experiment NOBC are displayed in Figure 8 (b) and (d). Figure 8 (b) shows that EXP-2 correction scheme effectively improves the temperature analysis fields accuracy in the upper and near-surface levels. EXP-2 scheme shows a better improvement than EXP-GSI schemes, especially in the upper levels. For the water vapor analysis fields (Figure 8 (d)), all BC schemes showed less significant improvements relative to NOBC compared to the temperature analysis fields, with changes in RMSE within 0.5%. It is still the EXP-2 scheme reducing the largest RMSE at 400 to 800 hPa atmosphere. This result proves that the bias correction scheme can enhance the assimilation effect of HIRAS-II data.

For details, please see Line 397-417. In addition, the website for conventional data used in data assimilation experiments is added in "Data availability" (Line 453).

**Minor comments**

Abstract: There are no explanations about the word EXP-2 in the abstract.

Re: The explanation about the word EXP-2 is added in Line 17.

L5, L26, L46, L64, L72, L88, L93, L98, L104, L105, L119, L120, L170, L178, L203, L222, L297, L337, L412: radiation -> radiance. The word radiance is usually used to mean measured quantity. In some contexts, the words brightness temperature may be more appropriate.

Re: The revision has been completed.

Fig.8: The figures should show confidence range by error bars.

Re: The error bars are added in Figure.8 (b) and (d).

**Other points:**

L91: but not provide --> but do not provide

Re: The revision has been completed.

L133: with a \*solar/viewing\* zenith angle?

Re: It is with a viewing zenith angle. The revision has been completed.

L339: WRFDA model \*has been\* developed

Re: The revision has been completed.

Thank you for your professional suggestions and concern about our manuscript again.